# Population receptive fields in nonhuman primates from whole-brain fMRI and large-scale neurophysiology in visual cortex

P Christiaan Klink[1,2]*, Xing Chen[1], Wim Vanduffel[3,4,5,6], Pieter R Roelfsema[1,2,7]

[1]Netherlands Institute for Neuroscience, Royal Netherlands Academy of Arts and Sciences, Amsterdam, Netherlands; [2]Psychiatry Department, Amsterdam UMC, Amsterdam, Netherlands; [3]Laboratory for Neuro- and Psychophysiology, Department of Neurosciences, KU Leuven Medical School, Leuven, Belgium; [4]Massachusetts General Hospital, Martinos Ctr. for Biomedical Imaging, Charlestown, United States; [5]Leuven Brain Institute, KU Leuven, Leuven, Belgium; [6]Harvard Medical School, Boston, United States; [7]Department of Integrative Neurophysiology, Center for Neurogenomics and Cognitive Research, VU University, Amsterdam, Netherlands

*For correspondence:
c.klink@nin.knaw.nl

Competing interest: The authors declare that no competing interests exist.

**Abstract** Population receptive field (pRF) modeling is a popular fMRI method to map the retinotopic organization of the human brain. While fMRI-based pRF maps are qualitatively similar to invasively recorded single-cell receptive fields in animals, it remains unclear what neuronal signal they represent. We addressed this question in awake nonhuman primates comparing whole-brain fMRI and large-scale neurophysiological recordings in areas V1 and V4 of the visual cortex. We examined the fits of several pRF models based on the fMRI blood-oxygen-level-dependent (BOLD) signal, multi-unit spiking activity (MUA), and local field potential (LFP) power in different frequency bands. We found that pRFs derived from BOLD-fMRI were most similar to MUA-pRFs in V1 and V4, while pRFs based on LFP gamma power also gave a good approximation. fMRI-based pRFs thus reliably reflect neuronal receptive field properties in the primate brain. In addition to our results in V1 and V4, the whole-brain fMRI measurements revealed retinotopic tuning in many other cortical and subcortical areas with a consistent increase in pRF size with increasing eccentricity, as well as a retinotopically specific deactivation of default mode network nodes similar to previous observations in humans.

## Editor's evaluation

This study is a detailed, systematic comparison of visually evoked population receptive fields (pRFs) measured non-invasively with MRI and invasively with electrophysiology in the same primate species. The authors show that MRI pRFs provide a good estimate of receptive fields based on multi-unit spiking activity in early visual areas. These results make an important contribution to our understanding of human imaging data in research and in the clinic.

## Introduction

The concept of a receptive field (RF) is crucial for our understanding of the mechanisms underlying perception, cognition, and action. RFs (*Hartline, 1938*; *Sherrington, 1906*) typically describe stimulus locations that evoke or modulate neuronal responses, but they can be generalized to different

stimulus features such as color or spatial frequency. RFs are usually measured by determining the neuronal firing rate elicited by visual stimuli (*Hubel and Wiesel, 1998*; *Hubel and Wiesel, 1968*; *Hubel and Wiesel, 1959*), but they can also be defined based on other neuronal signals such as subthreshold activity (*Priebe, 2008*), properties of the local field potential (LFP; *Victor et al., 1994*), or calcium levels that can, for instance, be measured with fluorescent calcium indicators (*van Beest et al., 2021*; *Bonin et al., 2011*).

Noninvasive methods lack the spatial resolution to measure the RF properties of single neurons, but they can characterize the RF properties of the aggregate neural signals being measured. The retinotopic organization of the human brain has now been characterized with functional magnetic resonance imaging for decades (*Wandell et al., 2007*; *Wandell and Winawer, 2011*). Early studies used phase-encoding with 'rotating wedge' and 'expanding or contracting ring' stimuli to identify RF position (*Engel, 2012*; *Engel et al., 1994*; *Sereno et al., 1995*), while later studies increasingly used the 'population receptive field' (pRF) method that estimates RF size in addition to position (*Dumoulin and Wandell, 2008*; *Wandell et al., 2007*; *Wandell and Winawer, 2015*; *Wandell and Winawer, 2011*). The method is popular and has been used to map a range of visual and cognitive functions (*Binda et al., 2018*; *Ekman et al., 2020*; *Harvey et al., 2020*; *Harvey et al., 2015*; *He et al., 2019*; *Hughes et al., 2019*; *Mo et al., 2018*; *Poltoratski et al., 2019*; *Poltoratski and Tong, 2020*; *Puckett et al., 2020*; *Shao et al., 2013*; *Shen et al., 2020*; *Silson et al., 2018*; *Stoll et al., 2020*; *Thomas et al., 2015*; *Welbourne et al., 2018*; *Zuiderbaan et al., 2017*), dysfunctions (*Ahmadi et al., 2020*; *Alvarez et al., 2020*; *de Best et al., 2019*; *Dumoulin and Knapen, 2018*; *Green et al., 2019*; *Schwarzkopf et al., 2014*), mechanisms of brain development (*Dekker et al., 2019*), cortical evolution (*Keliris et al., 2019*; *Kolster et al., 2014*; *Zhu and Vanduffel, 2019*), and information transfer across different brain areas (*Haak et al., 2013*).

The term 'pRF' highlights the analogy to neuronal RFs. It assumes that the blood-oxygen-level-dependent (BOLD) signal measured with fMRI reflects the aggregate response of a large population of neurons within a voxel. Indeed, pRFs of the human visual cortex are qualitatively similar to the RFs of single neurons or multi-unit activity (MUA) in animals (*Dumoulin and Wandell, 2008*). However, most of the previous comparisons were between species, and between studies that used different techniques to measure pRFs/RFs (*Barlow et al., 1966*; *Hubel and Wiesel, 1968*). Exceptions are pRF studies based on intracranial recordings in human patients (*Harvey et al., 2013*; *Winawer et al., 2013*), which used a limited number of surface electrodes to measure ECoG or intracranial EEG, but not spiking activity. The pRFs derived from the LFP exhibited similar properties to pRFs derived from BOLD signals, including similar spatial summation characteristics (*Winawer et al., 2013*). Another study compared BOLD-based pRFs in monkeys to the RF properties of single units in published work (*Kolster et al., 2014*). Furthermore, *Keliris et al., 2019* found that single-unit RFs in one of their monkeys were smaller than pRFs measured with BOLD and proposed another method to estimate RF sizes. Their study included MUA but not the LFP. A systematic within-species comparison of pRFs derived from BOLD, MUA, and LFP has however never been carried out.

Here, we fill that gap with extensive pRF modeling based on BOLD, MUA, and LFP signals in macaque monkeys. The question that neuronal signal forms the basis of the fMRI-BOLD signal has far-reaching consequences for the interpretation of human neuroimaging results and is therefore a topic of ongoing debate and rigorous investigation (*Arthurs and Boniface, 2002*; *Bartels et al., 2008*; *Boynton, 2011*; *Drew, 2019*; *Ekstrom, 2010*; *Goense and Logothetis, 2008*; *Logothetis, 2010*; *Logothetis, 2003*; *Logothetis et al., 2001*; *Logothetis and Wandell, 2004*; *Maier et al., 2008*; *Schölvinck et al., 2010*; *Sirotin and Das, 2009*; *Winawer et al., 2013*; *Winder et al., 2017*). Some studies reported that properties of the BOLD signal resemble features of both neuronal spiking and the LFP (*Mukamel et al., 2005*; *Nir et al., 2007*; *Rees et al., 2000*), others that they resemble the LFP but not spiking (*Bartolo et al., 2011*; *Maier et al., 2008*; *Niessing et al., 2005*; *Viswanathan and Freeman, 2007*), and yet others that they resemble spiking rather than the LFP (*Lima et al., 2014*). Here, we examine the degree to which pRFs based on the BOLD signal resemble pRFs based on spiking activity and distinct frequency bands of the LFP (*Buzsáki, 2006*; *Buzsáki and Draguhn, 2004*; *Einevoll et al., 2013*; *van Kerkoerle et al., 2014*). In nonhuman primates, we measured BOLD-pRFs using whole-brain fMRI and determined neuronal pRFs with large-scale neurophysiological recordings in V1 and V4 (*Figure 1*). Besides showing the presence of retinotopic information throughout the brain based on the fMRI data, we could directly compare V1 and V4 pRFs obtained with fMRI

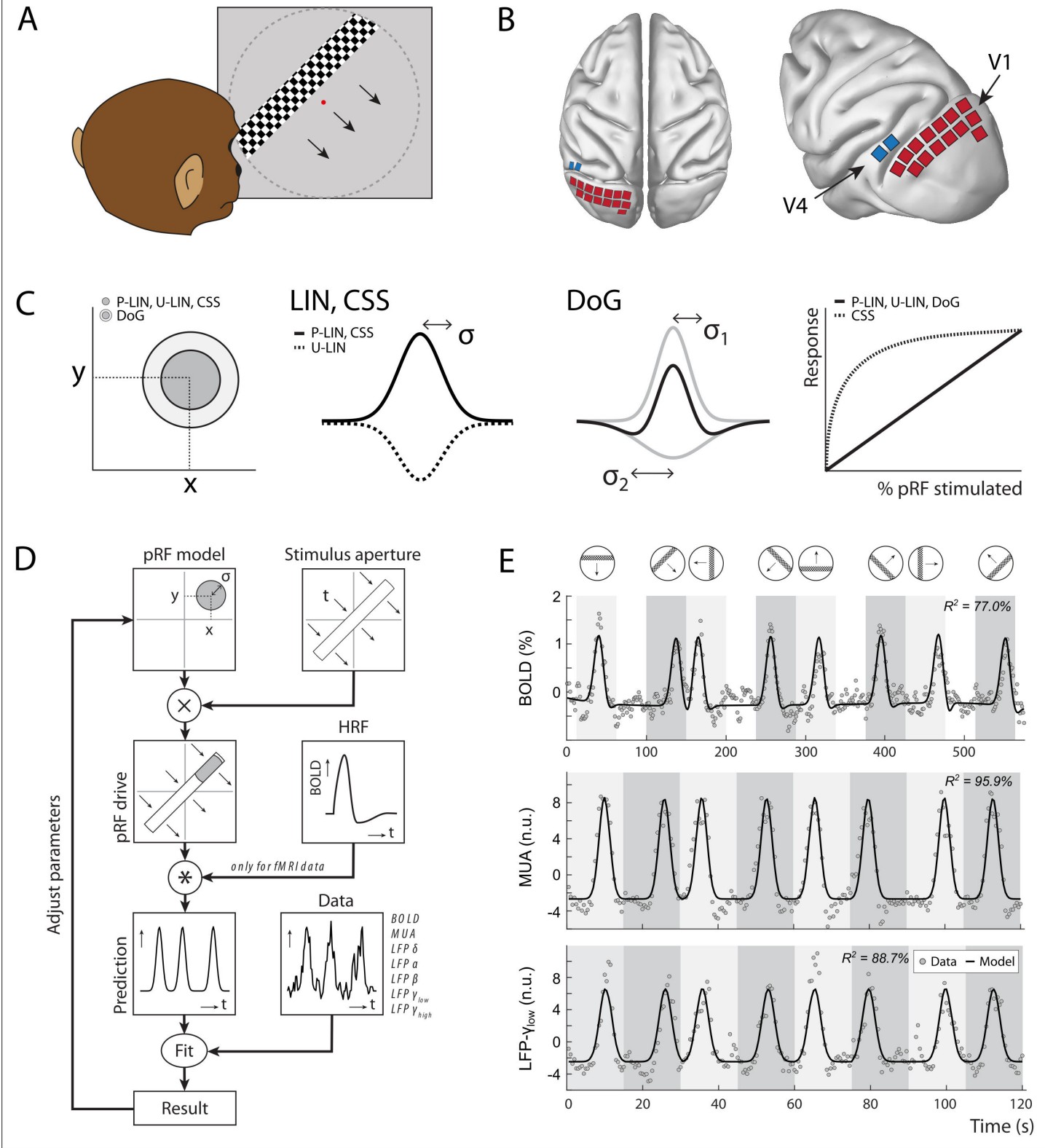

**Figure 1.** Experimental setup and study design. (**A**) Monkeys maintained fixation on a red dot while bars with high-contrast-moving checkerboards moved across the screen in eight different directions behind a virtual aperture (dashed line, not visible in the real stimulus). (**B**) Two animals performed the task in the MRI scanner. Two other animals were each implanted with 16 Utah arrays (1024 electrodes/animal) in the left visual cortex. The approximate locations of 14 V1 arrays (red) and 2 V4 arrays (blue) for one animal are depicted on the NMT standard macaque brain. For more detailed array configurations, see *Chen et al., 2020*. (**C**) Four population receptive field (pRF) models were fit to the data, differing in their pRF parameters

*Figure 1 continued on next page*

*Figure 1 continued*

(location: x, y; size: σ) and spatial summation characteristics. The difference-of-Gaussians (DoG) pRFs are described by an excitatory center and an inhibitory surround (both 2D Gaussians with $\sigma_1$ and $\sigma_2$ as size parameters; dark and light gray circles in the left panel, respectively). Other models fitted single Gaussians that were either constrained to be positive (second panel: solid line) or allowed to be negative (unconstrained linear U-LIN, dashed line). The compressive spatial summation (CSS) model implemented nonlinear spatial summation across the receptive field (RF) (fourth panel: dashed line), while all other models implemented linear summation (solid line). (**D**) The pRF model fitting procedure. A model pRF is multiplied with an 'aperture version' of the bar stimulus to generate a predicted response. For fMRI data, this prediction was convolved with a monkey-specific hemodynamic response function (HRF). The difference between the recorded neural signal (blood-oxygen-level-dependent [BOLD], multi-unit activity [MUA], local field potential [LFP]) and the predicted response was minimized by adjusting the pRF model parameters. (**E**) Examples of data and model fits for a V1 voxel (top panel) and a V1 electrode (middle and bottom panels). Average activity (gray data points) depicts the BOLD signal (top), MUA (middle), and LFP power in the low gamma band (bottom). Black lines are the model fits for a P-LIN pRF model. Light and dark gray areas depict visual stimulation periods (bar sweeps as indicated by the icons above). In the white epochs, the animals viewed a uniform gray background to allow the BOLD signal to return to baseline (these epochs were not necessary in the electrophysiology recordings). Note that that in MRI trials the bar stimuli had a lower speed (1 TR or 2.5 s per stimulus location) than during electrophysiology (500 ms per stimulus location).

The online version of this article includes the following figure supplement(s) for figure 1:

**Figure supplement 1.** Comparison of hemodynamic response functions (HRFs).

and electrophysiology. This intraspecies comparison provides new insight into the neurophysiological basis of the BOLD-defined pRFs and offers a benchmark for visual field maps obtained with fMRI.

The original method of estimating pRFs from BOLD responses (*Dumoulin and Wandell, 2008*) uses a forward modeling approach to fit the location and size of a symmetrical two-dimensional Gaussian to the BOLD responses. This approach is often used to predict neuronal activity elicited by moving bar-shaped stimuli. The RF model minimizes the difference between measured and predicted responses by multiplying the pRF with the stimulus and convolving the result with a hemodynamic response function (HRF), which accounts for the time course of neurovascular coupling (*Figure 1*). Later refinements implemented a difference-of-Gaussians pRF profile (DoG) to account for center-surround interactions (*Zuiderbaan et al., 2012*; *Figure 1C*), with substantial improvements in early visual cortex. Another refinement is the introduction of a static nonlinearity that models nonlinear spatial summation across RFs (*Britten and Heuer, 1999*; *Kay et al., 2013*; *Oleksiak et al., 2011*; *Winawer et al., 2013*). In such a model, the best parameters indicate subadditive spatial summation in all visual areas (*Kay et al., 2013*). This means that if stimulus $S_1$ elicits a response $R_1$ and a nonoverlapping stimulus $S_2$ elicits response $R_2$, the response to the combined stimulus, $S_1 + S_2$, is smaller than the sum, $R_1 + R_2$. For this reason, the nonlinear spatial summation model has also been called the 'compressive spatial summation' (CSS) model. A third extension has been the modeling of negative pRFs. Standard approaches tend to only include voxels that show increases in the BOLD signal in response to a stimulus. The inclusion of 'negative' pRFs with decreased BOLD activity has revealed the retinotopic organization of a number of areas in the so-called default mode network (DMN) (*Szinte and Knapen, 2020*). In our analysis of pRFs based on BOLD, MUA, and LFPs, we explored several pRF models, allowing us to investigate the potential presence of nonlinear spatial summation and negative pRFs.

## Results

Four macaque monkeys (*Macaca mulatta*) participated in this study. They were rewarded with fluid for maintaining their gaze inside a 1.5° window around a fixation point that was presented at the center of a frontoparallel screen. While they fixated, a 2° wide bar containing full-contrast moving checkerboards traversed the screen in eight different directions (*Figure 1*). Two animals performed this task in a 3T horizontal bore MRI scanner. Two other monkeys were each implanted with 1024 electrodes in the visual cortex (V1, V4). They performed the same task while neuronal activity (MUA and LFP) was recorded simultaneously from all electrodes. In the MRI setup, the stimulus covered 16° of the visual field, which was the maximum possible with the monitor located just outside of the scanner bore. The bar traveled across this screen in 20 steps of 2.5 s (1 TR). In the electrophysiology setup, the monitor was closer to the animal, allowing a visual field coverage of 28°. The stimulus bar moved across this aperture in 30 steps of 500 ms. For both the MRI and electrophysiology recordings, we only included data from epochs when the animals maintained fixation for >80% of the time.

After preprocessing (see Materials and methods), we independently fit four pRF models to the average BOLD time courses. These models were (1) a linear pRF model constrained to have positive

responses (P-LIN) (*Dumoulin and Wandell, 2008*), (2) an unconstrained version of the linear pRF model that can also model negative responses (U-LIN), (3) a DoG pRF model (*Zuiderbaan et al., 2012*), and (4) a nonlinear CSS pRF model (*Kay et al., 2013*; *Figure 1*). We used the fitting method to determine pRF size, shape, and location. A cross-validated goodness of fit ($R^2$) was determined by fitting the model to one half of the data and calculating $R^2$ using the other half of the data. Cross-validation allows the comparison of fit quality between models with different numbers of parameters.

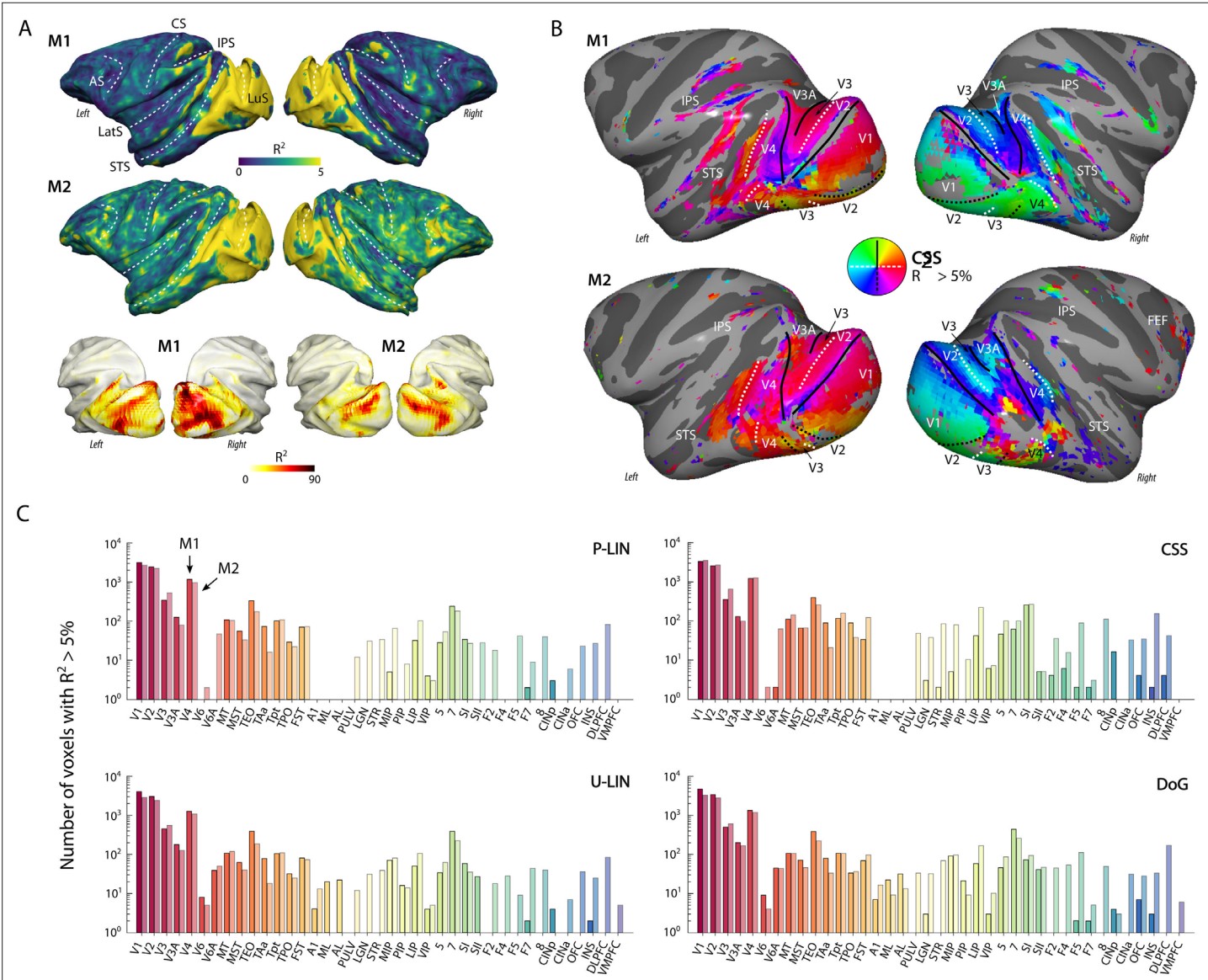

**Figure 2.** Population receptive field (pRF) model fits and retinotopic maps. (**A**) $R^2$ value map of the compressive spatial summation (CSS) pRF model projected on the surface rendering of the brains of two monkeys (M1, M2). The lower panel illustrates that the $R^2$ value in the visual cortex is generally much higher than 5% (going up to ~90%). The color range in the upper panels also reveals the weaker retinotopic information elsewhere in the cortex. AS, arcuate sulcus; CS, central sulcus; IPS, intraparietal sulcus; LatS, lateral sulcus; LuS, lunate sulcus; STS, superior temporal sulcus. (**B**) Polar angle maps for both subjects from the CSS model, thresholded at $R^2 > 5\%$, displayed on the inflated cortical surfaces. Functional delineation of several visual areas is superimposed. FEF, frontal eye fields. (**C**) Number of (resampled) voxels with $R^2 > 5\%$ per brain area and subject for each model. Note the logarithmic scale. See *Figure 2—figure supplement 1* for proportions per region of interest (ROI), and *Supplementary file 1* for a table with all ROI abbreviations.

The online version of this article includes the following figure supplement(s) for figure 2:

**Figure supplement 1.** Proportion of voxels with $R^2 > 5\%$ per region of interest (ROI).

## pRFs measured with BOLD-fMRI

All models provided good fits to the BOLD time courses in a range of cortical and subcortical areas known to be involved in visual processing. For both monkeys (M1 and M2), we found robust retino-topic information in occipital, temporal, and parietal cortex (*Figure 2*). pRFs in all these areas were in the contralateral visual field and retinotopic maps were consistent with previous reports (*Figure 2B*), some of which were more extensive (*Arcaro et al., 2011*; *Arcaro and Livingstone, 2017*; *Brewer et al., 2002*; *Janssens et al., 2014*; *Kolster et al., 2014*; *Rima et al., 2020*; *Zhu and Vanduffel, 2019*). Weaker and sparser retinotopic information was also observed in the frontal cortex, for example, around the arcuate sulcus (area 8, including the frontal eye fields [FEF]) and in the ventro-lateral prefrontal cortex (VLPFC). Throughout this study, we will use a voxel inclusion criterion of $R^2 >$ 5% unless otherwise noted. While $R^2$ was generally much higher in visual areas (*Figure 2A*, bottom panel), this relatively low threshold also reveals retinotopic information in more frontal areas and some subcortical regions (where the signal picked up by surface coils has a lower signal-to-noise ratio [SNR]). *Figure 2C* shows the number of voxels within a range of areas for which the models explained more than 5% of the variance (*Figure 2—figure supplement 1* shows the proportion per region of interest [ROI]). The functional parcellation of visual areas based on field sign inversions around hori-zontal and vertical meridians lined up well with a probabilistic atlas, co-registered to the individual animal's anatomy (D99, *Reveley et al., 2017*).

Subcortically, we could segregate the lateral geniculate nucleus (LGN), pulvinar and some striatal regions from their surrounding areas on the basis of a higher $R^2$. In both monkeys, the LGN of both hemispheres contained clear retinotopic maps (*Figure 3A*). Retinotopic information was also evident in the bilateral pulvinar of M1, but some of the pRFs in the pulvinar of M2 were large and crossed the vertical meridian, resulting in noisy polar angle maps (*Figure 3B*). Striatal retinotopy was less pronounced in M1 and even more variable in terms of polar angle maps (*Figure 3—figure supple-ment 1*).

We calculated cross-validated $R^2$ values to compare the four pRF models: P-LIN, U-LIN, DoG, and CSS (*Figure 1C*). A comparison of model performance pooled over subjects and voxels confirmed that there were significant differences across models, with the CSS model outperforming the P-LIN model (Kruskal–Wallis test on all four models, H = 21.33, df = 3, p<0.0001; post-hoc Tukey's HSD multiple comparisons of mean rank, $R^2_{CSS} > R^2_{P\text{-}LIN}$, p<0.0001). Indeed, the CSS model fit was better than that of P-LIN in all ROIs in M1 and in 38 out of 39 ROIs in M2 (Wilcoxon signed-rank, p<0.05; no difference in premotor area F7 in M2) (*Figure 4*). Since the CSS model also provided the best fits for the neurophysiological signals (described later), we report results from this model and only extend the analysis to other models where this is useful (e.g., in the case of negative gain values). The advantage of the CSS model over the P-LIN model generally increased in higher visual areas (*Figure 4—figure supplement 1*).

The static nonlinearity parameter, or 'pRF exponent' that models the nonlinearity of spatial summa-tion (Materials and methods: *Equation 4*), was in the range of 0.2–0.4 and significantly below 1 in all areas (Wilcoxon signed-rank, one-tailed, all ROIs with more than four voxels $R^2 > 5\%$, p<0.001). These values of the pRF exponent represent subadditive (compressive) spatial summation, in accordance with observations in the human visual cortex (*Kay et al., 2013*; *Winawer et al., 2013*). The pRF expo-nent in early visual cortex is comparable to previously reported values for human V1. The exponent value in higher visual areas was similar to that in early visual cortex of the two monkeys, and higher than previously observed in human extrastriate cortex, which suggests that spatial suppression is less pronounced in higher areas of the monkey visual cortex than in higher areas of the human visual cortex.

Both the U-LIN model and the DoG model also performed better than the standard P-LIN model (Kruskal–Wallis, Tukey's HSD, both ps<0.0001). The DoG had slightly better fits across all pooled voxels than the U-LIN model (Kruskal–Wallis, Tukey's HSD, p<0.0001). The advantage of the DoG model over the P-LIN model was most pronounced in V1 and decreased in higher visual areas.

## Negative pRFs from suppressed BOLD responses

There was a subset of voxels with negative BOLD responses for which both the U-LIN and DoG models provided much better pRF fits than the P-LIN and CSS models (arrows in *Figure 4A*, *Figure 4—figure supplement 1C*). We inspected the voxels for which the $R^2$ in the U-LIN/DOG models was at least

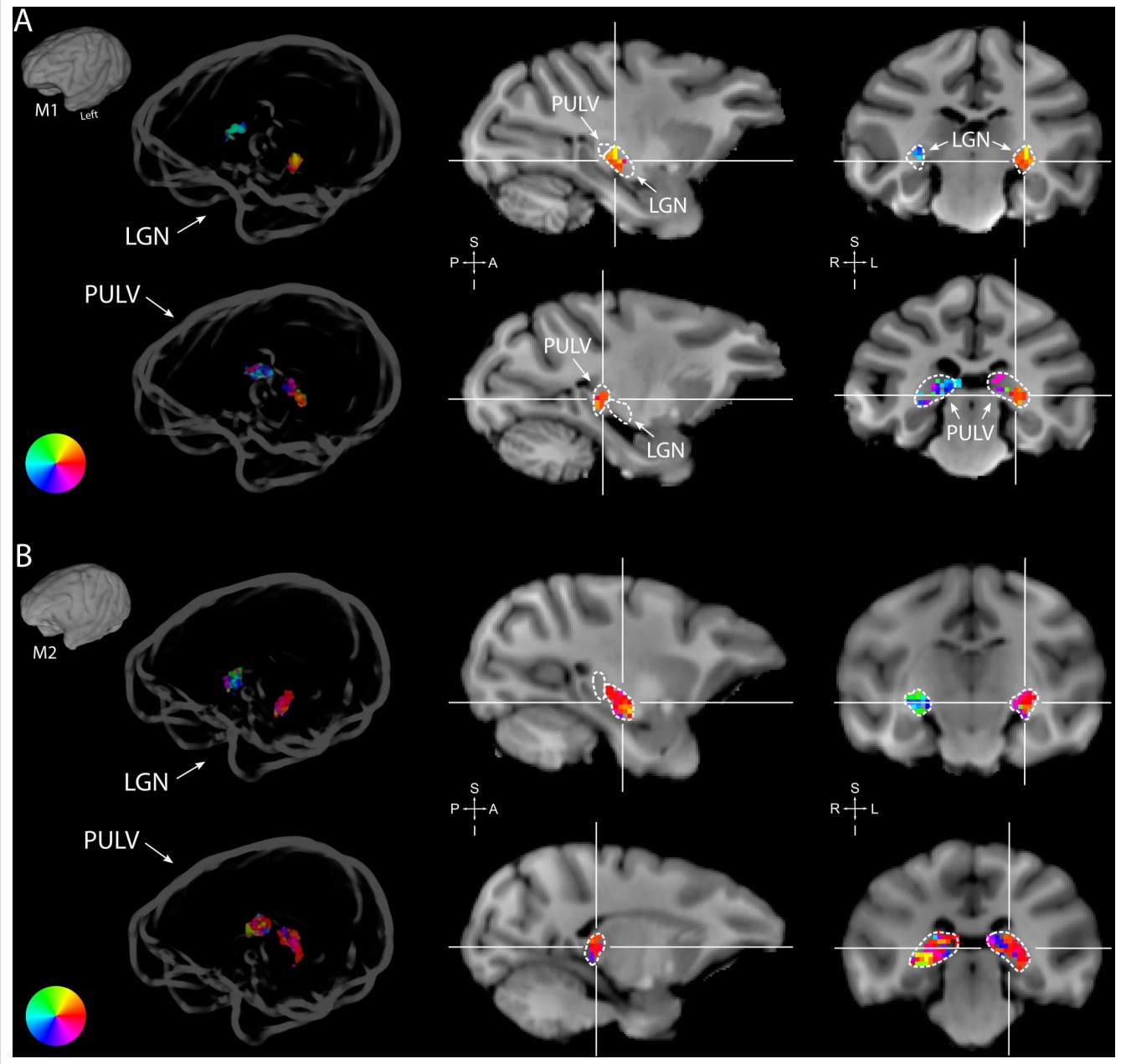

**Figure 3.** Retinotopy in the thalamus. Thalamic population receptive fields (pRFs) in M1 (**A**) and M2 (**B**). The lateral geniculate nucleus (LGN, top rows) contained retinotopic maps of the contralateral visual field in both monkeys (M1: 38/38; M2: 73/80 voxels with contralateral pRFs). Retinotopic information was also present in the pulvinar (PULV, bottom rows), but its organization was much less structured, especially in M2 (M1: 23/32; M2: 61/131 voxels with contralateral pRFs). Voxels were thresholded at $R^2 > 3\%$ for these polar angle visualizations due to the generally poorer fits in subcortex compared to visual cortex. Results from the compressive spatial summation (CSS) model are masked by region of interest (ROI) and shown both in a 'glass' representation of the individual animals' brains (left), and on selected sagittal and coronal slices (monkey-specific T1-weighted images, cross-hairs indicate slice positions). Dashed lines indicate the boundaries of the LGN and pulvinar.

The online version of this article includes the following figure supplement(s) for figure 3:

**Figure supplement 1.** Striatal population receptive fields (pRFs).

5% higher than in the P-LIN model (gray, dashed triangles in *Figure 4A*). The U-LIN model estimated a negative gain for these voxels (median gain = –0.31, Wilcoxon signed-rank, one-tailed, z = –43.9, p<0.0001) (*Figure 4B*) and the DoG model returned a high level of inhibition (median normalized suppressive amplitude = 1.14, interquartile range [IQR], 0.98–1.29).

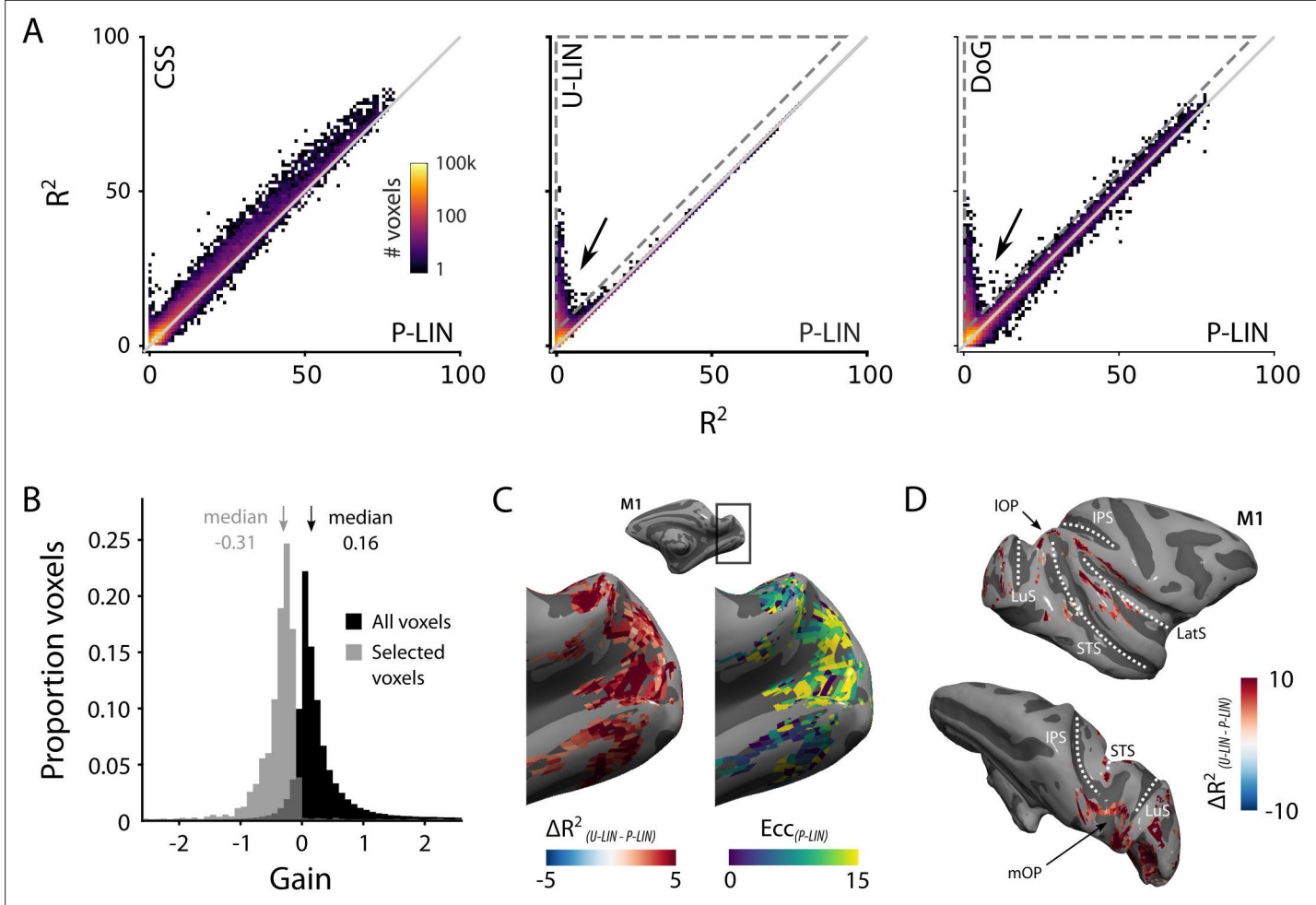

**Figure 4.** Comparison of the four population receptive field (pRF) models. (**A**) Comparison across pRF models. $R^2$ data are in bins of 1% × 1%, and color indicates the number of voxels per bin. The compressive spatial summation (CSS) model fits the data best, while U-LIN and difference-of-Gaussians (DoG) models give better fits for voxels that are poorly characterized by the P-LIN model (arrows). Dashed triangle, voxels for which $R^2$ of the U-LIN/ DoG models was at least 5% higher than that of the P-LIN model. (**B**) Gain values of the U-LIN pRF fits for all voxels (black) and the voxels in the gray triangle in (**A**.) The negative gain values indicate visual suppression of the blood-oxygen-level-dependent (BOLD) signal. Arrows, medians. (**C**) Clusters of voxels for which the U-LIN model fit better than the P-LIN model were located in the medial occipital lobe (left panel). The pRFs of these voxels had a high eccentricity according to the P-LIN model, beyond the stimulated visual field. (**D**) Clusters of voxels with negative pRFs in the U-LIN model (without positive pRFs in the P-LIN model) were present around the lateral sulcus (LatS), in the medial occipital parietal cortex (mOP) and in the lateral occipital parietal cortex (lOP).

The online version of this article includes the following figure supplement(s) for figure 4:

**Figure supplement 1.** Comparison of fit performance across pRF models.

**Figure supplement 2.** Location of negative pRFs.

There were two categories of voxels with negative responses (*Figure 4—figure supplement 2*). For the first category of voxels, the P-LIN model estimated pRFs outside the boundaries of the stimulated visual field. This result suggests that the negative response represents surround suppression that is particularly strong around the fovea (*Sereno et al., 1995*; *Shmuel et al., 2006*; *Smith et al., 2004*). The retinotopy of these voxels is consistent with this explanation. In V1, for instance, the voxels were on the medial side of the occipital pole, which represents the peripheral visual field (*Figure 4C*). The second category of voxels with negative pRFs appeared to be different. Here, P-LIN and CSS models could not fit any pRF, suggesting purely negative BOLD responses. These voxels were primarily located in the medial occipital parietal cortex (mOP), at the superior border of the superior temporal sulcus in the lateral occipital parietal cortex (lOP) and around the lateral sulcus (LatS), which includes parts of the insula, cingulate, parietal, and premotor cortices (*Figure 4D*). These areas have

all previously been identified as being part of the monkey DMN (*Mantini et al., 2011*). This finding aligns with recent research in humans that revealed similar negatively tuned pRFs in corresponding nodes of the human DMN (*Szinte and Knapen, 2020*).

## pRF size as a function of eccentricity

As expected, pRF sizes were larger at higher eccentricities. This relationship was evident in all areas with larger numbers of well-fit voxels. pRFs were also larger in higher areas, which exhibited a steeper slope of the eccentricity-size relationship (*Figure 5*, *Figure 5—figure supplements 1 and 2*). The differences between the slopes in V1 and V2 are smaller than expected based on previous electro-physiological studies, but this is not uncommon with fMRI (*Kay et al., 2013*). In one animal, we also unexpectedly observed retinotopy in a number of higher areas, such as the anterior cingulate cortex (*Figure 5—figure supplements 1 and 2*). This brain region has been studied predominantly in the context of decision-making (*Amiez et al., 2006*; *Fouragnan et al., 2019*), but it does have resting state correlations with V1 (*Griffis et al., 2017*). Our design lacked the power for a more detailed investigation of this retinotopic organization, but this result may inspire future work on brain-wide retinotopic tuning.

## Multi-unit spiking activity RFs

We determined the RFs of MUA recorded with chronically implanted electrode arrays (Utah arrays) in areas V1 and V4 in two additional monkeys (M3 and M4) that did not participate in the fMRI experiments. We used a 1024-channel cortical implant, consisting of a titanium pedestal that was connected to 16 Utah arrays (*Rousche and Normann, 1998*). Each Utah array had 8 × 8 shanks with a length of 1.5 mm. In both monkeys, 14 arrays were placed in V1 and 2 in V4 of the left hemisphere (*Figure 6*). The stimulus was similar to that used in the fMRI experiments with some small differences due to constraints of the two setups (in the electrophysiology setup, the stimulus covered a larger portion of the visual field because the screen was closer to the animal) and the intrinsic nature of the recorded signals (stimulus steps were faster in the electrophysiology experiments because the electrophysiology signals are much faster than the BOLD response). We fit the four pRF models to the MUA responses and to the LFP power in five distinct frequency bands. We compared the MUA pRFs to a more conventional MUA RF-mapping method (cRFs). For this cRF method, we selected channels with an SNR larger than 3 (i.e., visual responses that were more than three times larger than the standard deviation of the spontaneous activity) and derived the RF borders from the onset and offset of the neuronal activity elicited by a smoothly moving light bar (see Materials and methods). Whenever both methods were able to estimate a pRF and cRF ($R^2 > 25\%$ for the pRF method, SNR > 3 and $R^2 > 25\%$ for the cRF method), the estimated locations were highly similar (median distance between pRF and cRF center, V1: 0.34, IQR 0.18–0.49 dva; V4: 0.90, IQR 0.40–1.41 dva). Compared to the P-LIN pRF model, the moving bar method estimated smaller cRFs (median size difference $pRF_{sz}\text{-}cRF_{sz}$: 0.50, IQR: 0.08–0.92) (*Figure 7*). The CSS model, however, returned pRF size estimates that were very similar to the cRF sizes or even a little bit smaller (median size difference $pRF_{sz}\text{-}cRF_{sz}$: –0.13; IQR: –0.41–0.14), suggesting that nonlinear spatial summation might indeed be better at capturing the RF properties of a small population of neurons than linear summation.

The pRF models that were used in this study are all based on circular (symmetric) RFs. A recent study (*Silson et al., 2018*) suggested that pRFs in human early visual cortex might be elliptical rather than circular, although this suggestion goes against previous work (*Greene et al., 2014*; *Merkel et al., 2018*; *Zeidman et al., 2018*). A later study demonstrated that the elliptical fits were an artifact of the software that had been used in the analysis (*Lerma-Usabiaga et al., 2021*). The cRF method separately estimates the width and height of the RF, and can thus be used to calculate a simplified RF aspect ratio to investigate RF symmetry. While this measure differs from RF ellipticity (a 45° tilted ellipse has the same aspect ratio as a circle), it does provide some insight into the symmetry of the MUA RFs. We did observe a few cRFs with aspect ratios ($\sigma_{large}/\sigma_{small}$) that were larger than 2 (M3: 18/753; M4: 10/527; together 2.2% of all cRFS), but the vast majority of cRFs in both animals had aspect ratios close to 1 (M3 median: 1.12, IQR: 1.04–1.20; M4 median 1.13, IQR: 1.03–1.22) indicating near-symmetric RFs.

We obtained excellent fits to the MUA for all pRF models (see *Figure 1E* for an example fit). These pRFs covered a large proportion of the lower-right visual field (*Figure 6*, *Figure 6—figure*

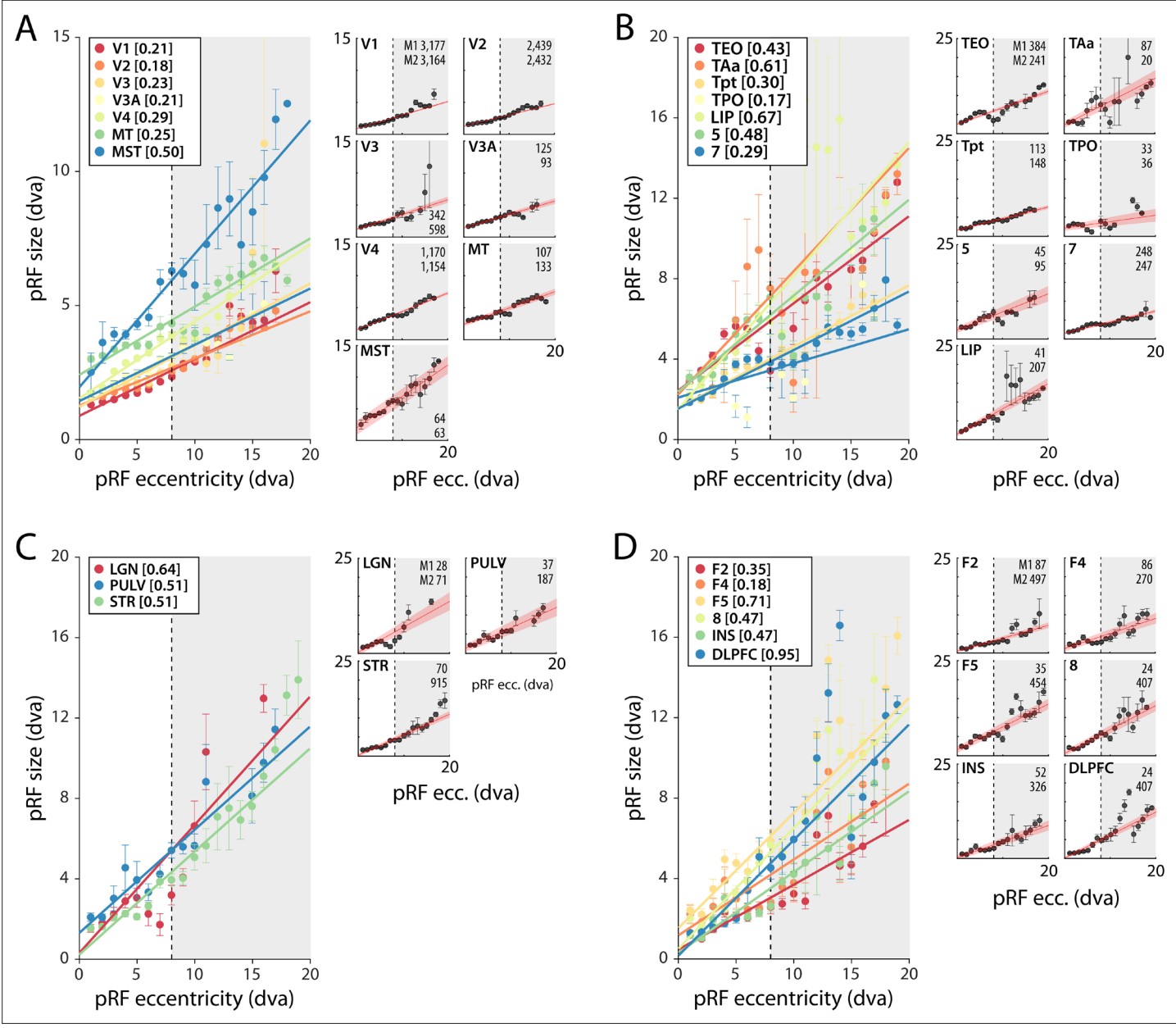

**Figure 5.** Population receptive field (pRF) size as a function of eccentricity according to the compressive spatial summation (CSS) model. (**A**) Eccentricity-size relationship for early and mid-level visual areas. (**B**) Eccentricity-size relationship for areas in the temporal and parietal lobes. (**C**) Eccentricity-size relationship for subcortical areas. (**D**) Eccentricity-size relationship for frontal cortical areas. Data points are pRF sizes binned in 2-dva-eccentricity bins; error bars denote SEM. Lines are linear fits with a significant slope (p<0.01). Slope values are shown between square brackets in the legend. The dashed line that separates the white and gray areas indicates the extent of the visual stimulus used to estimate pRFs. Data points in the gray regions come for pRFs that fell partially outside the region with a visual stimulus. The small panels show the same data for individual areas and include confidence intervals of the linear fit (shaded red area). Numbers denote the number of voxels per animal. The displayed areas were selected based on the presence of at least 20 voxels in each animal, with $R^2 > 5\%$ for (**A, B**) and $R^2 > 3\%$ for (**C, D**) due to the generally lower fit quality in the subcortex and frontal lobe. *Figure 5—figure supplements 1 and 2* plot all suprathreshold voxels.

The online version of this article includes the following figure supplement(s) for figure 5:

**Figure supplement 1.** Eccentricity-size relationship for all regions of interest (ROIs).

**Figure supplement 2.** Eccentricity-size relationship for all regions of interest (ROIs).

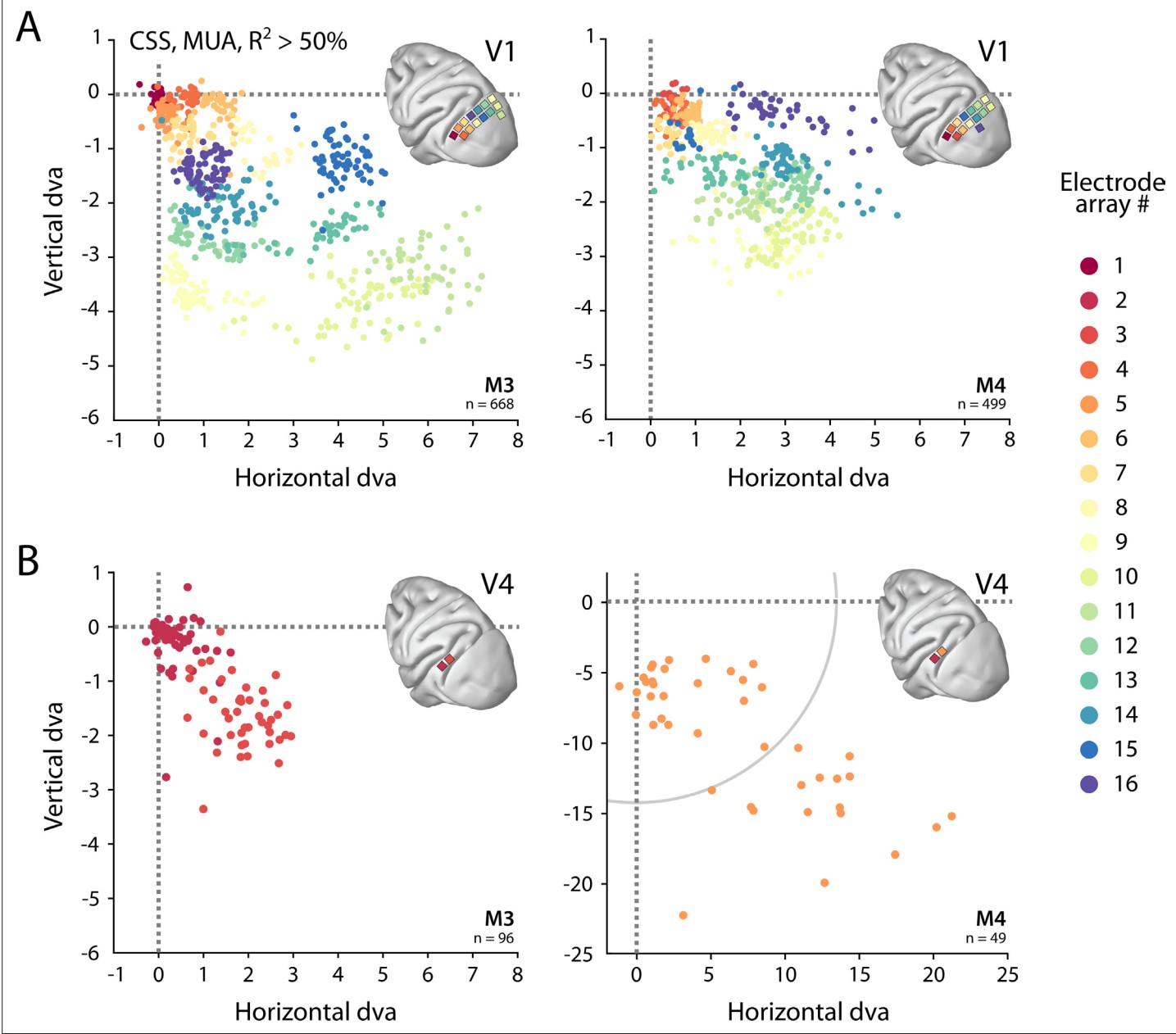

**Figure 6.** Visual field coverage of population receptive fields (pRFs) with Utah arrays. (**A**) In both monkeys (M3, M4), 14 Utah arrays were implanted on the left operculum that is partly V1. Different colors represent the center of multi-unit activity (MUA)-based pRFs for the individual arrays. Only electrodes with $R^2 > 50\%$ in the compressive spatial summation (CSS) model are shown. (**B**) Same as in (**A**), but for V4 electrodes. Note the different scale in the lower-right panel, with the gray arc indicating the extent of the visual stimulus. See *Figure 6—figure supplement 1* for pRF sizes.

The online version of this article includes the following figure supplement(s) for figure 6:

**Figure supplement 1.** Heatmaps of visual field coverage of the Utah arrays.

supplement 1). As expected, the pRFs from electrodes of the same arrays (shown in the same color in *Figure 6*) were clustered in space, and their locations were in accordance with established retinotopy (e.g., *Hubel and Wiesel, 1974*). The average $R^2$ (over all electrodes with $R^2 > 0$) was 64% in V1 (M3: 54%; M4: 73%) and 53% in V4 (M3: 37%; M4: 68%), which is substantially higher than the average $R^2$ of 11% in both V1 and V4 for the MRI data (all voxels with $R^2 > 0$; V1 M1: 14%, M2: 8%; V4 M1: 14%, M2: 8%).

Cross-validated comparisons revealed significant differences between the four models (Kruskal–Wallis test on all four models: $H_{V1} = 204$, $df_{V1} = 3$, $p_{V1} < 0.0001$; $H_{V4} = 13.4$, $df_{V4} = 3$, $p_{V4} < 0.01$)

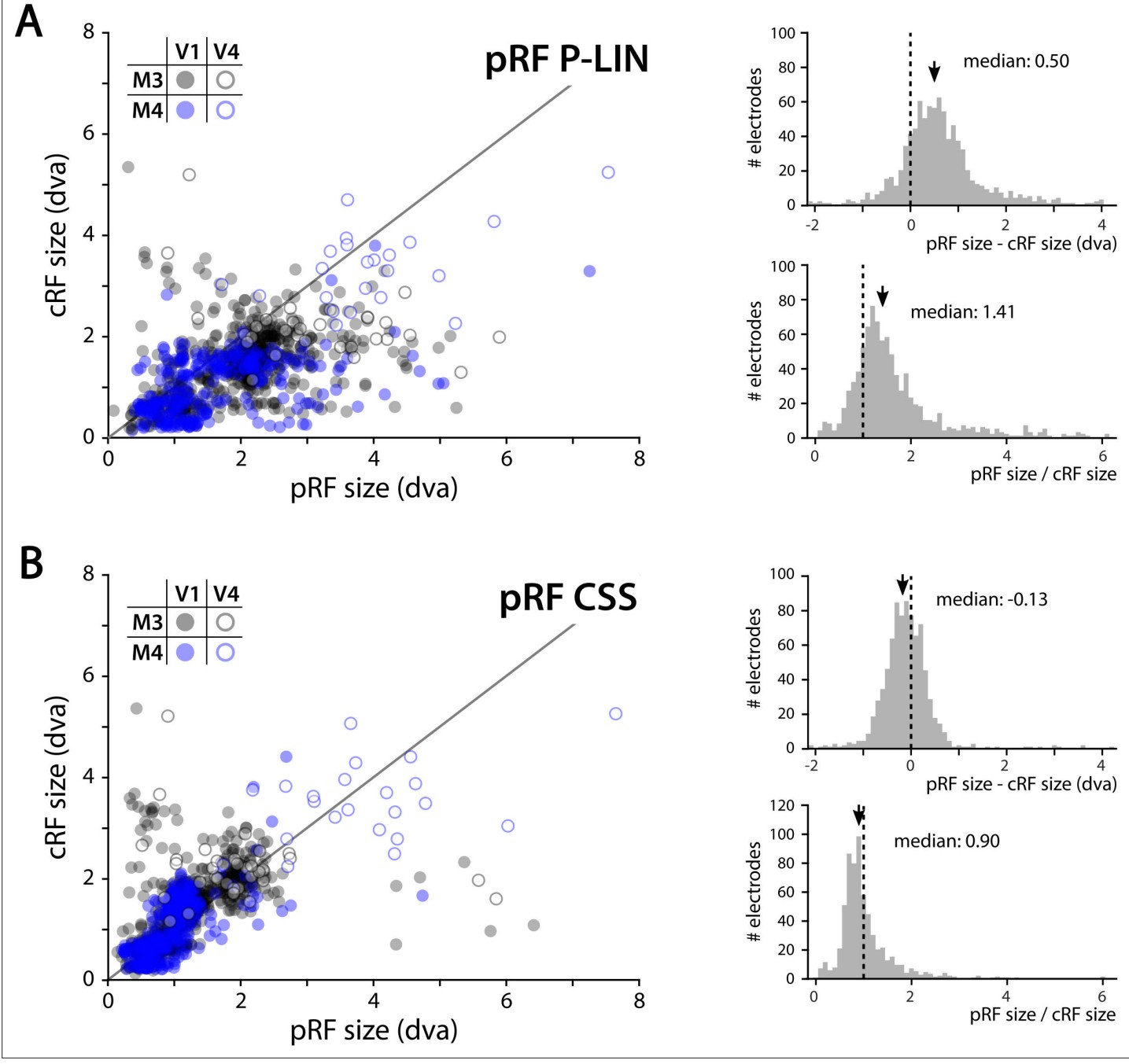

**Figure 7.** Comparison of multi-unit activity (MUA) population receptive field (pRF) sizes with conventionally determined RF (cRF) sizes (moving bar stimulus). Data points represent recording sites of individual animals (black: M3; blue: M4) and brain areas (closed circles: V1; open circles: V4). (**A**) pRF sizes estimated with the P-LIN model (X-axis in left panel) are larger than cRF sizes obtained with a thin moving luminance bar (Y-axis in left panel). The median difference between pRF and cRF sizes across all electrodes (pooled across animals and areas) was 0.50 (interquartile range [IQR]: 0.08–0.92) and the median ratio was 1.41 (IQR: 0.99–1.83), as shown in the top and bottom-right panels, respectively. (**B**) As in (**A**), but for pRF sizes estimated with the compressive spatial summation (CSS) model, which are slightly smaller than the cRFs (median difference: –0.13, IQR: –0.41–0.14; median ratio 0.90, IQR: 0.67–1.12).

(*Figure 8*; similar patterns were present in each individual animal). Post-hoc pairwise comparisons (Tukey's HSD) revealed that the CSS and DoG models provided a better fit than the linear models, although for V4 the advantage of the CSS model over the P-LIN model was only significant when electrodes with a poor fit ($R^2 < 25\%$) were excluded (V1, all electrodes: CSS vs. P-LIN, p<0.001, DoG

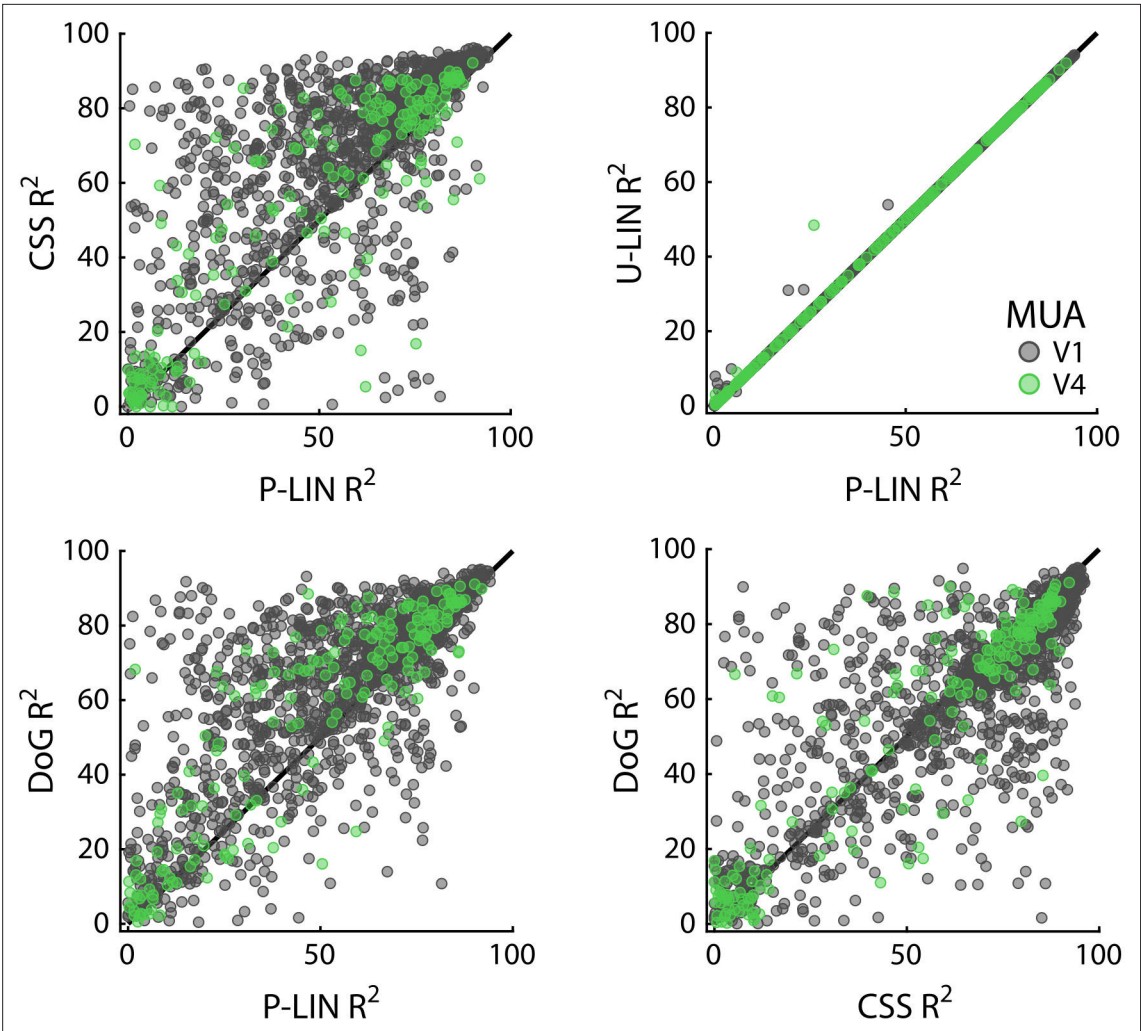

**Figure 8.** Comparison of multi-unit activity (MUA)-based fit results from the four population receptive field (pRF) models. Scatterplots compare $R^2$ of pRF models. Each dot represents an electrode (black: V1; green: V4).

The online version of this article includes the following figure supplement(s) for figure 8:

**Figure supplement 1.** Comparison of local field potential (LFP) fits for the four population receptive field (pRF) models in V1.

**Figure supplement 2.** Comparison of local field potential (LFP)-based fit results from the four population receptive field (pRF) models in V4.

**Figure supplement 3.** Comparison of population receptive field (pRF) fit accuracies for multi-unit activity (MUA) and local field potential (LFP) signals at the same recording sites.

vs. P-LIN, p<0.001; V4, all electrodes: CSS vs. P-LIN, p=0.16, DoG vs. P-LIN, p<0.001; V1, electrodes with $R^2$ > 25%: CSS vs. P-LIN, p<0.001, DoG vs. P-LIN, p<0.001; V4, electrodes with $R^2$ > 25%: CSS vs. P-LIN, p<0.02, DoG vs. P-LIN, p<0.01) (*Figure 8*). The improved fit of the DoG model was caused by the suppressive surround (median normalized suppressive amplitude = 0.71, IQR 0.56–0.86).

The pRF exponent of the CSS model was 0.38 ± 0.23 in V1 (mean ± SD; 1358 electrodes with $R^2$ > 25%), which was significantly smaller than 1 (Wilcoxon signed-rank, one-tailed, z = –31.59, p<0.0001) and similar to the MRI-based values, which had a mean of 0.34 ± 0.19 (6341 voxels). Likewise, in V4, MUA pRF exponent values were significantly smaller than 1 (0.33 ± 0.19; z = 11.13, p<0.0001; n = 165), and comparable to MRI-based values in the same area (0.30 ± 0.16; n = 2324). The similarity in the values of the pRF exponent indicates that subadditive spatial summation is a prominent feature in these areas.

The size of the estimated MUA pRFs increased with eccentricity (*Figure 9*). However, the pRF sizes for two of the V1 arrays in both monkeys were approximately three times smaller than expected when compared to the data from the other arrays. These outlying arrays were located on the posterior-medial

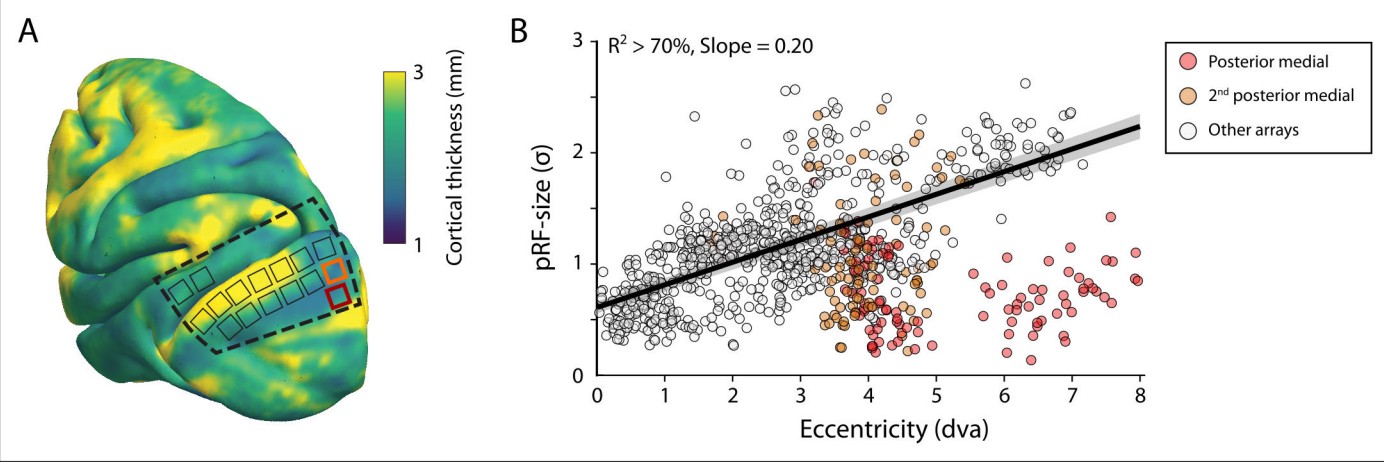

**Figure 9.** V1 arrays with outlying population receptive field (pRF) sizes. (**A**) Schematic representation of the location of the craniotomy made during surgery (dashed line) and the implanted electrode arrays (rectangles) depicted on the NMT standard brain. The color map indicates the thickness of the cortical gray matter of the NMT. (**B**) For both monkeys, the estimated pRF sizes of V1 electrode arrays in the posterior medial corner of the craniotomy (red and orange data points correspond to red and orange rectangles in panel **A**) were surprisingly small for their eccentricity compared to the size-eccentricity relationship seen in the other arrays (gray circles, linear fit with 95% CI as black line and gray area). Given the length of the electrodes (1.5 mm), the typical thickness of the striate cortex, and these small pRF sizes, it is likely that the outlying pRFs do not reflect the tuning of V1 neurons, but that of the geniculostriate pathway in the white matter. The pRF sizes were estimated by the compressive spatial summation (CSS) model (recording sites shown have $R^2 > 70\%$).

side of the surface of V1 in both monkeys where the gray matter is relatively thin (**Figure 9A**). We therefore suspect that the 1.5-mm-long shanks of the Utah arrays were pushed into the white matter, where they picked up activity of thalamic afferents (i.e., the geniculostriate pathway). We therefore excluded these electrodes from the analyses of pRF sizes. The remaining MUA pRF sizes and eccentricities were highly similar to RFs reported in previous electrophysiology studies at similar eccentricities (**Gattass et al., 1987**; **Gattass et al., 1981**; **Van Essen et al., 1984**; **Victor et al., 1994**). Next, we compared pRFs between the electrophysiological signals and the fMRI-BOLD signal.

## Local field potential pRFs

The LFP was split into five frequency bands: $\theta$ (4–8 Hz), $\alpha$ (8–16 Hz), $\beta$ (16–30 Hz), $\gamma_{low}$ (30–60 Hz), and $\gamma_{high}$ (60–120 Hz). We fit the four models to estimate pRFs for each frequency band. The results for $\gamma_{low}$ and $\gamma_{high}$ resembled those for the MUA with high $R^2$ values for a large proportion of the electrodes (especially in V1). The CSS model again outperformed the other models, and we did not observe negative pRFs in these V1 and V4 regions (**Figure 8—figure supplements 1 and 2**). Electrodes with good MUA-pRF fits usually also had good LFP-pRFs, but the opposite was not always true (**Figure 8—figure supplement 3**). The pRFs in lower frequency bands differed. Whereas $\theta$ generally yielded low $R^2$ values, $\alpha$ and $\beta$ yielded good fits for a substantial number of electrodes. Interestingly, there were two classes of electrodes in these frequency bands. For the first class, the power increased with visual stimulation and CSS model fits were best. In contrast, the second class of electrodes had negative pRFs, that is, the stimulus suppressed power (**Figure 8—figure supplement 1**). Given the few electrodes with good low-frequency LFP-pRFs in V4, we focused our analysis on the positive and negative responses on V1 (the split into positive/negative pRF was based on the parameters of the U-LIN model; **Figure 10**, **Figure 10—figure supplement 1**).

We observed a number of remarkable differences between the positive and negative pRFs. First, negative pRFs generally had lower eccentricities than positive pRFs (**Figure 10B**, **Figure 10—figure supplement 1B**). Second, negative pRFs were larger than positive pRFs (**Figure 10C**, **Figure 10—figure supplement 1C**). Third, there was a systematic difference in the distance from the γ-pRF of the same electrode. Specifically, we calculated a 'separation index' by dividing the distance between the centers of the low- and high-frequency LFP-pRFs by their summed size estimates (SI = Distance/($\sigma_{lf}$ + $\sigma_{hf}$)). A separation index of less than 1 indicates pRF overlap. For both α and β power, the negative pRFs were farther from the γ-pRFs than the positive pRFs (**Figure 10D**, **Figure 10—figure supplement 1D**)

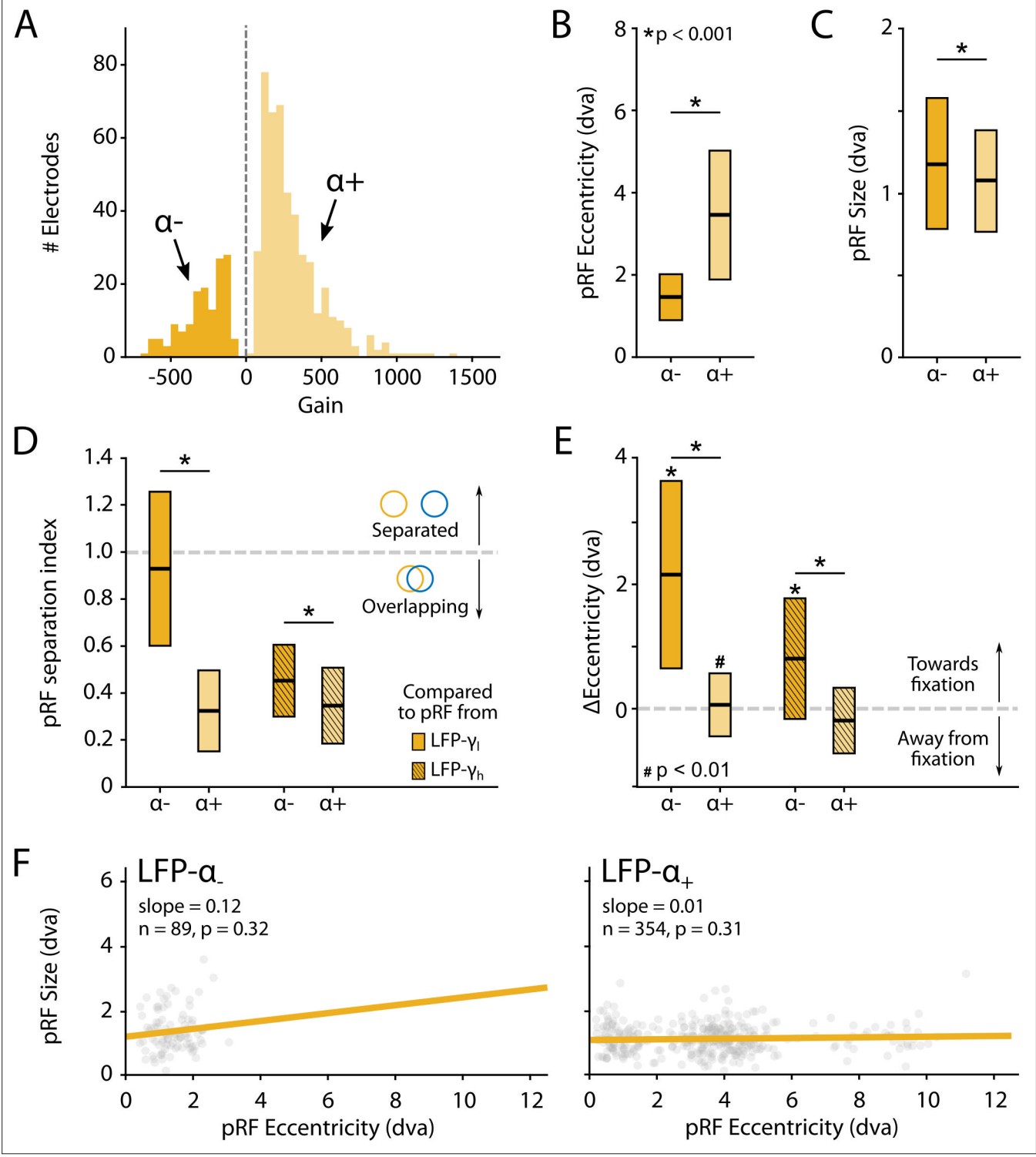

**Figure 10.** Characteristics of local field potential (LFP)-α population receptive fields (pRFs) in V1 split by positive and negative gain values. (**A**) Distribution of gain values for LFP-α pRFs of V1 electrodes estimated with the U-LIN model. Electrodes with positive gain pRFs are classified as α+, electrodes with negative gain pRFs as α-. (**B, C**) pRF eccentricities (**B**) and size (**C**) for α- and α+ electrodes (yellow shades). Colored boxes indicate interquartile range (IQR), and the median is shown as a thick horizontal line. Wilcoxon rank-sum tests were used for comparisons between α- and α+ electrodes. (**D**) Distance between the centers of LFP-α and LFP-γ pRFs from the same electrode, divided by the sum of their respective sizes. Values smaller than 1 indicate overlapping receptive fields. Nonshaded boxes are comparisons with LFP-γ low pRFs, shaded boxes are comparisons with LFP-γ high pRFs. (**E**) Difference in eccentricity between LFP-γ and LFP-α pRFs from the same electrodes (calculated as Ecc_γ -Ecc_α). Positive values indicate

*Figure 10 continued on next page*

*Figure 10 continued*

that LFP-α pRFs are closer to fixation than LFP- γ pRFs. Wilcoxon signed-rank, one-tailed test (ΔEcc > 0), for individual cases; Wilcoxon rank-sum test for comparisons between α- and α+ electrodes. See *Figure 10—figure supplement 2* for a visualization of the shifts per recording site. (**F**) Eccentricity-size relationship for α- (left) and α+ (right) electrodes. Dots indicate individual electrodes. *Figure 10—figure supplement 1* shows the same pattern of results for the LFP-β pRFs.

The online version of this article includes the following figure supplement(s) for figure 10:

**Figure supplement 1.** Characteristics of local field potential (LFP)-β population receptive fields (pRFs) in V1 split by positive and negative gain values.

**Figure supplement 2.** Negative population receptive fields (pRFs) based on low-frequency local field potential (LFP) components are shifted toward the fixation point compared to the positive pRFs based on the high-frequency LFP at the same recording site.

and generally shifted in the direction of fixation (*Figure 10E*, *Figure 10—figure supplements 1E and 2*). These results suggest that the positive pRFs represent visually driven activity, whereas the negative pRFs signal a form of suppression that is strongest at smaller eccentricities, close to the fixation point. One possible explanation is that the monkeys directed attention to the fixation point, which may have caused a ring of suppression closely surrounding it. Another possibility is that the negative pRFs might be a consequence of small eye movements around the fixation point. The size of α-pRFs did not depend on eccentricity (*Figure 10F*), whereas the size of β-pRFs increased with eccentricity, although this relation was very weak for positive pRFs (*Figure 10—figure supplement 1F*).

The pRFs derived from all electrophysiology signals on the same electrode had similar locations, with less than 1 dva between their centers on average (CSS model, *Figure 11A*). We next analyzed RF sizes, normalizing the size estimates to the MUA-cRF. All LFP-pRFs estimates were larger than MUA-pRFs, and lower frequency LFP components yielded larger pRFs than the higher frequencies (*Figure 11B*; we observed the same pattern present in each animal). The pRF exponent was well below 1 for all LFP components (V1 and V4 electrodes with $R^2 > 25\%$; Wilcoxon signed-rank, one-tailed <1, p<0.001) and smaller at lower frequencies, indicative of stronger CSS. We also compared the exponents to those of the MRI-pRFs. The exponent of the MRI was not significantly different from that of $\gamma_{low}$ in V1, and both $\gamma_{low}$ and $\gamma_{high}$ in V4 (p>0.05, *Figure 11C*). Differences between the MRI exponent and those of the other LFP bands were significant (V1: Kruskal–Wallis, H = 505.49, df = 6, p<0.0001; V4: Kruskal–Wallis, H = 21.65, df = 3, p<0.001; Tukey's HSD for multiple comparisons).

## Comparison of pRF eccentricity-size relationship between fMRI and electrophysiology signals

We next compared the eccentricity-size relationship of the electrophysiological signals to that of the BOLD-fMRI pRFs using linear mixed models (LMMs) to evaluate the pRF estimates of the CSS model, separately for V1 and V4 (electrodes with $R^2 > 50\%$ and $R^2 > 5\%$ for MRI). We included voxels for which the pRF was in the lower-right visual quadrant where we had electrode coverage. In V1, positive eccentricity-size relationships existed for MRI, MUA, β, $\gamma_{low}$, and $\gamma_{high}$ (*Figure 12A*). We compared these signals with a single LMM, revealing an interaction between signal type and eccentricity (F = 14.44, df = 4, p<0.0001), which indicated that the slopes differed. We further used pairwise LMMs to compare the slope of the MRI-pRFs to the electrophysiological signals with a significant eccentricity-size relationship. The fMRI eccentricity-size relationship was similar to that of MUA (F = 0.38, df = 1, p=0.54), whereas it was significantly different from all LFP signals (β: F = 7.97, df = 1, p<0.01; $\gamma_{low}$, F = 20.02, df = 1, p<0.001; $\gamma_{high}$, F = 20.05, df = 1, p<0.001) (*Figure 12C*). We repeated this analysis in V4 (*Figure 12B*) where we had fewer electrodes. We obtained positive eccentricity-size slopes for MUA, $\gamma_{low}$, and $\gamma_{high}$ but did not obtain fits of sufficient quality for the $\theta$, α, and β bands. In contrast to V1, there was no clear difference across these signals (F = 1.11, df = 3, p=0.24). Pairwise comparisons with the MRI size-eccentricity relationship did not reveal a difference between those for $\gamma_{low}$ (F = 0.36, df = 1, p=0.55), $\gamma_{high}$ (F = 0.28, df = 1, p=0.60), or MUA (F = 2.40, df = 1, p=0.12). Because the visual field coverage in V4 differed between the two animals that were used for electrophysiology, we repeated the analysis in both individuals and observed similar results (not shown).

To further investigate the robustness of these results, we repeated the analysis (1) with the inclusion of either all V1 and V4 voxels or a subset of the voxels in approximately the same anatomical location as the electrode arrays and (2) by varying the $R^2$-based inclusion criteria for electrodes and voxels

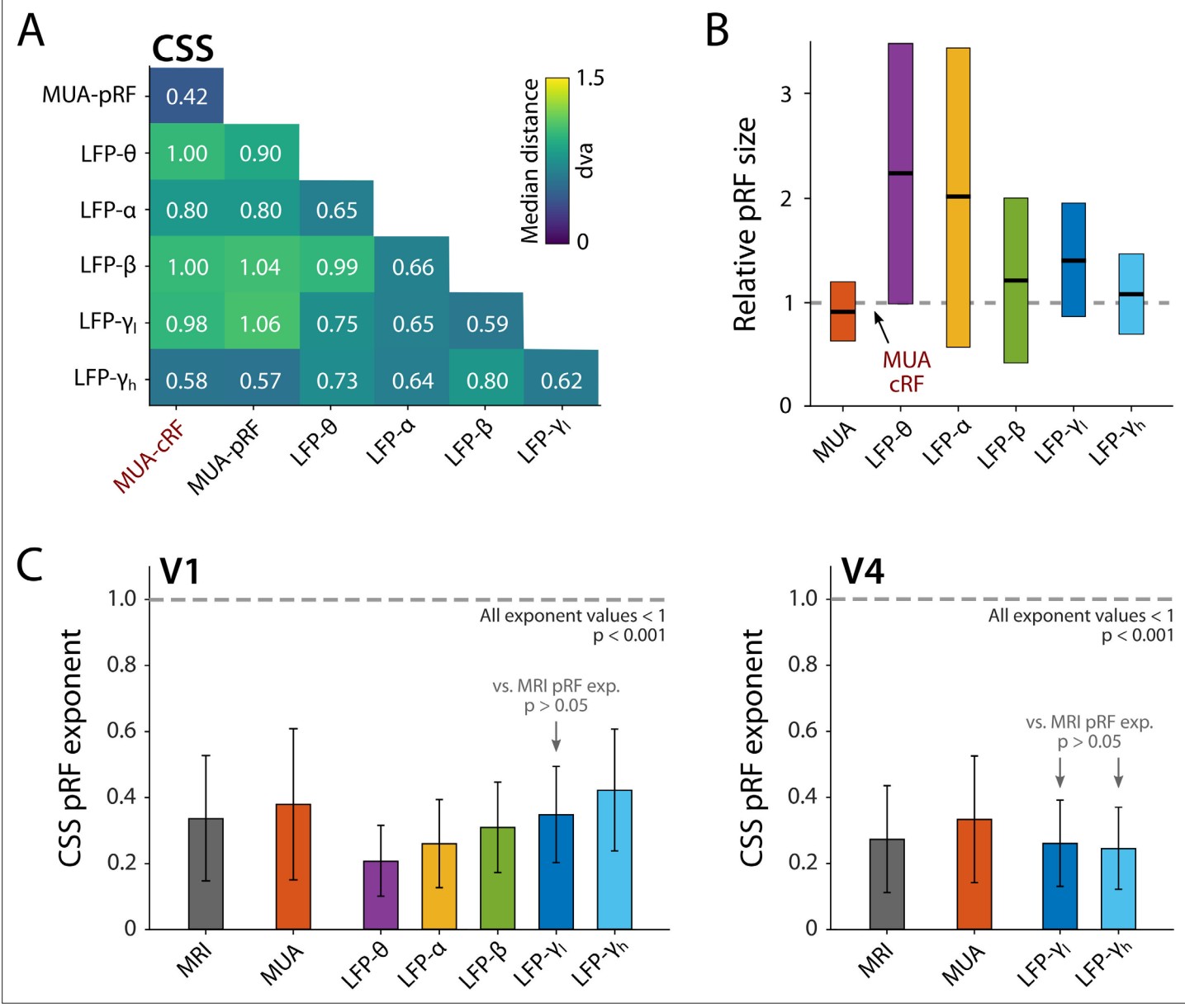

**Figure 11.** Comparison of population receptive field (pRF) location and size of different electrophysiological signals at the same electrode, estimated by the compressive spatial summation (CSS) model. (**A**) Median distance between receptive field (RF) estimates. Electrodes were only included if $R^2 >$ 25% (multi-unit activity [MUA], local field potential [LFP]) or signal-to-noise ratio (SNR) > 3 (MUA-cRF). (**B**) RF size for the electrodes of (**A**), normalized to the MUA-cRF (dashed line). Horizontal lines indicate the median, colored rectangles depict the interquartile range (IQR). (**C**) pRF exponent from the CSS model (indicating nonlinearity of spatial summation). The horizontal dashed line indicates linear summation. The exponent was significantly lower than 1 for all signals. Gray arrows indicate signals for which the exponent did not significantly differ from that of the MRI pRFs. A similar pattern was present in the electrophysiological data of each animal.

(**Figure 12—figure supplement 1**). MUA-pRFs in V1 and V4 were generally similar to the BOLD-pRFs, although in V4 $\gamma_{low}$ and $\gamma_{high}$ also approximated the fMRI results in some of the comparisons.

## Discussion

The current study conducted a systematic comparison of pRFs obtained with fMRI and electrophysiological recordings in V1 and V4 of awake behaving macaque monkeys. Within the same species, we fit several pRF models to seven different signal types (BOLD, MUA, and the power in five LFP frequency bands) to gain insight into the neuronal basis of MRI-BOLD-pRF measurements (**Wandell**

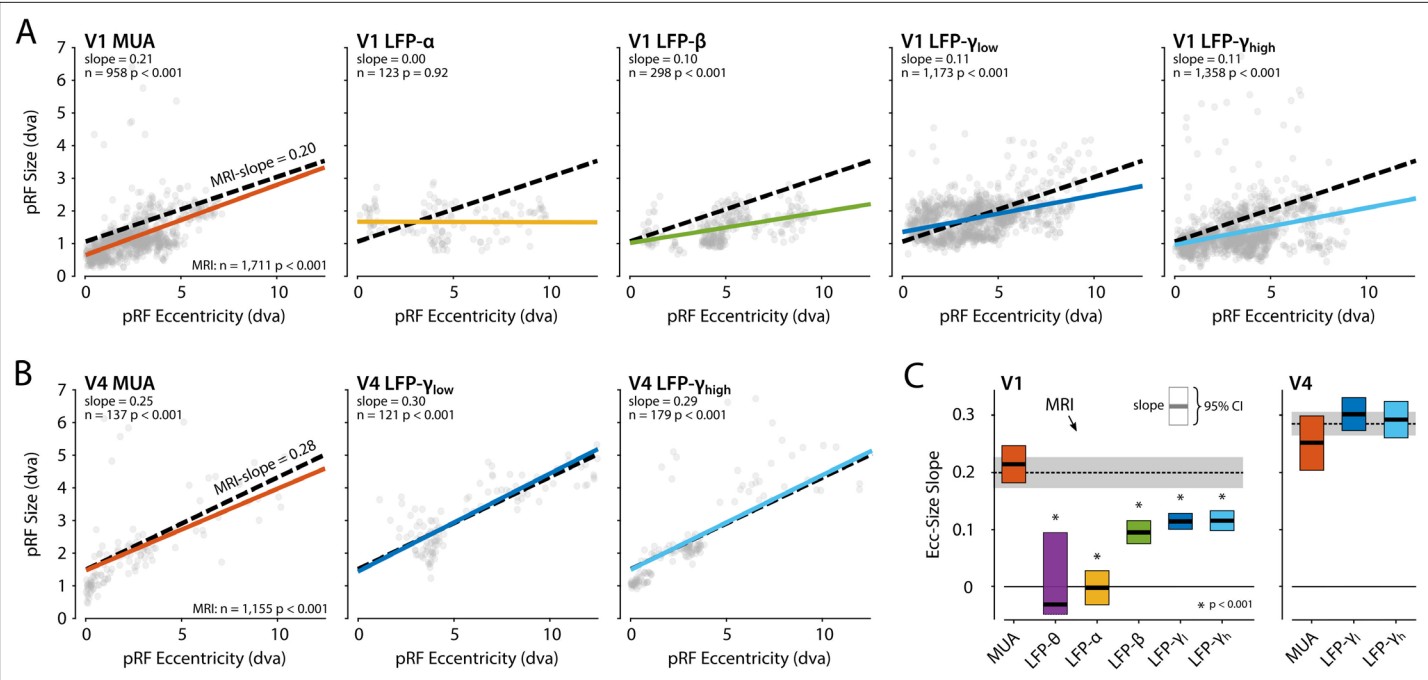

**Figure 12.** Eccentricity-size relationship for population receptive fields (pRFs) across signal types. (**A**) The pRF size-eccentricity relation for V1 electrodes (compressive spatial summation [CSS] model, $R^2 > 50\%$). Dots are individual electrodes, colored lines represent the slope of the eccentricity-size relationship. The dashed black line represents the relationship for V1 MRI voxels with a pRF in the lower-right visual quadrant and an $R^2 > 5\%$. We only included signals with >25 electrodes meeting the $R^2$ threshold. (**B**) Same as in (**A**), but now for the V4 electrodes and voxels. (**C**) Eccentricity-size slopes (left: V1; right: V4). The dashed line represents the slope for MRI-based pRFs with the 95% confidence interval depicted in gray shading. Colored rectangles indicate the 95% confidence intervals for the electrophysiological signals and the horizontal black line the slope estimate. The lower bound of the 95% confidence interval of the LFP-$\theta$ is not visible. Asterisks indicate significant difference with the size-eccentricity relation of the MRI-based pRFs.

The online version of this article includes the following figure supplement(s) for figure 12:

**Figure supplement 1.** Cross-signal comparisons of population receptive field (pRF) eccentricity-size relationship.

*and Winawer, 2015*). Our results demonstrate retinotopic tuning in many brain regions and the presence of negative pRFs in areas of the DMN. We found that subadditive spatial summation is a prominent feature of many measures of brain activity. Furthermore, the results establish a clear relationship between BOLD-pRFs and electrophysiologically determined pRFs, with MUA-pRFs being most similar to BOLD-pRFs (as discussed below).

## Cortical and subcortical retinotopic tuning of the BOLD signal

Retinotopic information was present in occipital, temporal, parietal, and frontal cortex as well as in a subcortical areas. We could differentiate the LGN and pulvinar from their surrounding areas based on their higher $R^2$. Retinotopic maps in the LGN were roughly in line with previously published retinotopic organization of the macaque LGN (*Erwin et al., 1999*), but they comprised few voxels and the variability across animals prohibited detailed inferences. In the pulvinar, the organization of the retinotopic information was even more variable across animals. In humans, subcortical retinotopic maps have been observed in LGN, pulvinar, superior colliculus (SC), thalamic reticular nucleus (TRN), and substantia nigra (SN) (*Cotton and Smith, 2007*; *DeSimone et al., 2015*; *Schneider et al., 2004*). Here, we also observed voxels with pRFs located lateral and medial to the pulvinar, but more targeted investigations are necessary for detailed individual-subject segmentation of the thalamus (*DeSimone et al., 2015*; *Tani et al., 2011*).

We obtained good pRF fits in the occipital, temporal, and parietal cortical areas, regions for which previous studies demonstrated retinotopic organization with phase-encoded retinotopic mapping (*Arcaro et al., 2011*; *Janssens et al., 2014*; *Kolster et al., 2014*; *Kolster et al., 2010*; *Kolster et al., 2009*; *Patel et al., 2010*). Despite the fact that checkerboard stimuli may not be ideal for the

activation of frontal areas, which are better driven by more complex stimuli (*Janssens et al., 2014*; *Saygin and Sereno, 2008*), we also observed retinotopy in several frontal areas, including the insula, cingulate cortex, FEFs (area 8), orbitofrontal cortex, ventromedial prefrontal cortex, and dorsolateral prefrontal cortex. These areas are thought to be involved in visual processing and visual attention. Retinotopic maps have previously been reported in the cerebellum in humans (*van Es et al., 2019*). We did not detect such maps, likely because of a lower SNR in the cerebellum due to lower field strength, a less optimal coil placement, and the sphinx position of the monkeys.

## Negative pRFs

We observed negative pRFs in the MRI data and in the low-frequency LFP, but they were of a different nature. We identified two classes of negative pRFs in the MRI data. Negative pRFs in the visual cortex were usually accompanied by positive responses at peripheral visual field locations, close the boundaries of the visual display. Similar negative visual BOLD responses have been reported in human visual cortex (*Smith et al., 2004*) and are presumably caused by surround suppression (*Allman et al., 1985*; *Cavanaugh et al., 2002*; *Hubel and Wiesel, 1962*; *Knierim and van Essen, 1992*). We found a second class of negative pRFs around the LatS, in the mOP, and at the superior border of the superior temporal sulcus in the IOP, all areas previously implicated in the monkey's DMN (*Mantini et al., 2011*). A similar retinotopy of negative responses was recently found in the human DMN (*Szinte and Knapen, 2020*). The retinotopy of nodes of the DMN in humans and monkeys implies a sensory-based organization of this part of the DMN, which could play a role in visual cognition (*Arsenault et al., 2018*).

Negative pRFs were also observed for a subset of the electrodes in the α and β range of the LFP, although pRF fits were generally of lower quality than those for the γ-LFP or MUA signals. It is conceivable that these negative pRFS reflect shifts of attention, which are known to modulate α and β power (*Griffiths et al., 2019*; *Siegel et al., 2008*; *Womelsdorf and Fries, 2007*; *Worden et al., 2000*). In accordance with this idea, negative α/β pRFs were larger than their positive counterparts and shifted towards the fixation point. Future studies are needed for a closer examination of this phenomenon.

## The relation between BOLD-based pRFs and electrophysiologically determined pRFs

We compared BOLD-based pRFs to electrophysiological measures in terms of their spatial summation characteristics, size, and the relationship between eccentricity and size (*Amano et al., 2009*; *Dumoulin and Wandell, 2008*; *Felleman and Van Essen, 1987*; *Gattass et al., 2005*; *Kay et al., 2013*; *Larsson and Heeger, 2006*; *Van Essen et al., 1984*; *Victor et al., 1994*). Eye movements can have an effect on the pRF size estimates, and it is therefore important to note that (1) we only included data from recordings where the animals maintained a high fixation performance (IQR-span of the horizontal and vertical eye position, M1: 0.23, 0.34 dva; M2: 0.36, 0.49 dva; M3: 0.18, 0.38 dva; M4: 0.37, 0.60 dva), and (2) we averaged across multiple stimulus presentations to obtain robust response profiles. Hence, variations in eye position can have had only minor effects on the present results.

## Compressive spatial summation

The CSS pRF model includes nonlinear spatial summation (*Kay et al., 2013*). Previous studies observed subadditive spatial summation, also known as spatial compression, throughout the human visual cortex in BOLD-based pRFs, with stronger compression in higher visual areas than in V1 (*Kay et al., 2013*; *Winawer et al., 2013*). In the present study, spatial compression was present in subcortical and cortical areas with good BOLD-pRF fits, indicating that it is a widespread, if not universal, characteristic of the primate visual system. The strength of compression did not differ much across areas, and it was similar to that in human V1. For the MUA data, the CSS model estimated pRF sizes that were very similar to RF size estimates derived from conventional methods that use moving luminance bars, whereas the P-LIN model systematically returned larger pRF estimates. This difference might indicate that spatial compression indeed better captures the neuronal RF properties, at least in V1 and V4.

In human iEEG recordings, spatial compression in the broadband iEEG signal was reported to be similar in strength to that of the BOLD signal (*Winawer et al., 2013*). We also observed CSS in both the MUA-pRFs and in all frequency bands of the LFP, and the pRF exponent tended to be smaller for lower-frequency components, indicative of stronger compression. The lower frequencies also

exhibited stronger suppression than the BOLD signal. In contrast, the pRF exponent of MUA and $\gamma_{low}$ was similar to that of BOLD-fMRI in both V1 and V4, but in V1 the exponent for $\gamma_{high}$ was larger. Hence, MUA and $\gamma_{low}$ were the two electrophysiological markers for which spatial compression resembled that of BOLD-fMRI most.

## pRF size and the eccentricity-size relationship

As expected, pRFs were larger at higher eccentricities, with a larger slope in higher visual cortical areas. The pRF sizes and the eccentricity-size relationships that we found in visual cortex were in line with previous results from human and monkey neuroimaging studies (*Amano et al., 2009*; *DeSimone et al., 2015*; *Dumoulin and Wandell, 2008*; *Kay et al., 2013*; *Keliris et al., 2019*; *Kolster et al., 2014*; *Kolster et al., 2010*; *Welbourne et al., 2018*; *Zhu and Vanduffel, 2019*; *Zuiderbaan et al., 2012*). Furthermore, the results are in keeping with previous electrophysiological recordings of single units and MUA in early visual areas such as V1 (*Gattass et al., 1987*; *Gattass et al., 1981*; *Van Essen et al., 1984*; *Victor et al., 1994*), V2 (*Burkhalter and Van Essen, 1986*; *Gattass et al., 1981*; *Rosa et al., 1988*), V3 (*Burkhalter and Van Essen, 1986*; *Felleman and Van Essen, 1987*; *Newsome et al., 1986*; *Rosa et al., 2000*), and V4 (*Gattass et al., 1988*). Finally, they also match the results of human electrophysiology with subdural electrodes (*Harvey et al., 2013*; *Yoshor et al., 2007*).

## The spatial scope of fMRI-BOLD, MUA, and LFP

The spatial scale of recorded neural signals depends both on the nature of the signal and the recording method. For extracellular electrophysiology recordings, the impedance and size of an electrode determine its sensitivity for single-unit spiking activity or MUA. This relationship is complex (*Viswam et al., 2019*; *Ward et al., 2009*), but high-impedance (~1 MΩ) electrodes are generally better suited for the detection of spiking activity, with smaller contact sites and higher impedance thought to sample from smaller neural populations and being more likely to pick up single-neuron activity. The spatial range of the MUA signal is ~140 μm (*Buzsáki and Draguhn, 2004*). Electrode impedance, size, and shape are less important for LFP recordings, at least within commonly used ranges (*Nelson and Pouget, 2010*; *Viswam et al., 2019*). The spatial extent of the LFP signal is a topic of ongoing debate (*Kajikawa and Schroeder, 2011*) with some authors estimating it as low as 120–250 μm in visual cortex (*Katzner et al., 2009*; *Xing et al., 2009*), while others suggest it may stretch up to several millimeters (*Berens et al., 2008*; *Katzner et al., 2009*; *Kreiman et al., 2006*), and may even be detected more than a centimeter away from the source (*Kajikawa and Schroeder, 2011*). The hypothesis that low-frequency components of the LFP have a larger spatial reach than high-frequency components is supported by computational modeling (*Leski et al., 2013*), although it has also been challenged (*Dubey and Ray, 2016*). Yet, the exact origin of the LFP and its relationship to spiking remain to be completely understood (*Einevoll et al., 2013*).

Given the larger spatial spread of the LFP compared to MUA, one would expect pRFs based on LFPs to be larger than those based on MUA. This is indeed what we found. The size of LFP-pRFs furthermore depended on the frequency component of the LFP, with lower frequencies yielding larger pRFs, suggesting a frequency-dependent spread of visual information in the LFP signal. In fMRI, there is a trade-off between spatial and temporal resolution. While higher magnetic field strengths and specialized acquisition methods are continuously increasing spatiotemporal resolution, studies with voxel sizes on the order of 1–2 mm isotropic and a repetition time of 2–3 s at 3T field strengths are still common. Due to the complex relationship between the hemodynamic BOLD signal and the underlying neural activity, it is difficult to predict its stimulus sensitivity from neuroimaging parameters such as voxel size. Nonetheless, a comparison of the eccentricity-size relationship between MRI and electrophysiology in V1 revealed that the BOLD-based pRFs resembled MUA in V1 and V4. In V4, however, the BOLD slope was also very similar to that of the LFP gamma power (*Figure 12—figure supplement 1*). One possible reason for this difference between areas is that V4 is smaller and more heterogeneous (*Kolster et al., 2014*; *Zhu and Vanduffel, 2019*) than V1. A V4 voxel; therefore, samples from a neuronal population with more heterogeneous spatial tuning than an equally sized V1 voxel may therefore reflect the activity of a larger population of neurons, which is better approximated by gamma power. Other factors that might play a role are potential differences in the quality of the recorded signal across areas or partial volume effects that are likely to be more prevalent in V4 than in V1.

## Conclusions

Our comparison of fMRI with large-scale neurophysiological recordings in visual cortex revealed that pRFs derived from the BOLD signal resemble MUA RFs. Subadditive spatial summation is a general feature of many brain areas and occurs for BOLD, MUA, and LFP. We observed negative pRFs in the monkey DMN and as part of center-surround organization of pRFs in early visual areas. The spatial compression and the eccentricity-size relationship of BOLD resembles that of MUA, but also bears a resemblance to gamma power. We conclude that BOLD-pRFs accurately represent the spatial tuning of the underlying neuronal populations.

# Materials and methods

**Key resources table**

| Reagent type (species) or resource | Designation | Source or reference | Identifiers | Additional information |
|---|---|---|---|---|
| Biological sample (*Macaca mulatta*) | Rhesus macaque (*Macaca mulatta*), male | Biomedical Primate Research Center, the Netherlands | n/a | - |
| Other | Philips Ingenia 3.0T MR system | Philips | n/a | At Spinoza Centre for Neuroimaging, Amsterdam, the Netherlands |
| Other | 8-channel phased array receive MR coil system | KU Leuven | n/a | Custom-built |
| Other | 16-channel MR pre-amplifier | MR Coils BV | n/a | Custom-built |
| Other | ETL-200 | ISCAN | RRID:SCR_021044 | MR-compatible eye tracker |
| Other | E3X-NH | Omron | n/a | Fiber optic amplifiers |
| Other | 5-RLD-E1 Liquid Reward System | Crist Instrument Company, Inc | n/a | Juice reward system |
| Other | BOLDscreen 32 LCD for fMRI | Cambridge Research Systems | n/a | MR-compatible display |
| Other | Utah array (electrodes) | Blackrock Microsystems | n/a | - |
| Other | 128-channel CerePlex M head-stages | Blackrock Microsystems | n/a | Data acquisition |
| Other | 128-channel CerePlex M head-stages | Blackrock Microsystems | n/a | Data acquisition |
| Other | 128-channel Digital Hub | Blackrock Microsystems | n/a | Data acquisition |
| Other | 128-channel Neural Signal Processor (NSP) | Blackrock Microsystems | n/a | Data acquisition |
| Software, algorithm | Blackrock Central Software Suite | Blackrock Microsystems | n/a | - |
| Other | ET-49C | Tomas Recording | n/a | Eye tracker |
| Software, algorithm | MATLAB | MathWorks | RRID:SCR_001622 | - |
| Other | LISA cluster | SURFsara | n/a | Computing cluster |
| Software, algorithm | dcm2niix | https://github.com/rordenlab/dcm2niix | RRID:SCR_01409 | - |
| Software, algorithm | Nipype | http://nipy.org/nipype/ | RRID:SCR_002502 | Used as the basis of the custom NHP-BIDS pipeline |

*Continued on next page*

*Continued*

| Reagent type (species) or resource | Designation | Source or reference | Identifiers | Additional information |
|---|---|---|---|---|
| Software, algorithm | NHP-BIDS | Netherlands Institute for Neuroscience | RRID:SCR_021813 | In-house developed, available via: https://github.com/VisionandCognition/NHP-BIDS |
| Software, algorithm | FreeSurfer | http://surfer.nmr.mgh.harvard.edu/ | RRID:SCR_001847 | Used as the basis of the custom NHP-Freesurfer |
| Software, algorithm | NHP-Freesurfer | Netherlands Institute for Neuroscience | RRID:SCR_021814 | In-house developed, available via: https://github.com/VisionandCognition/NHP-Freesurfer |
| Software, algorithm | Pycortex | https://gallantlab.github.io/pycortex/ | n/a | Used as the basis of the customized NHP-Pycortex |
| Software, algorithm | NHP-Pycortex | Netherlands Institute for Neuroscience | RRID:SCR_021815 | In-house developed, available via: https://github.com/VisionandCognition/NHP-pycortex |
| Software, algorithm | analyzePRF | https://kendrickkay.net/analyzePRF/ | n/a | Toolbox was edited for this study and made available with the code and data |
| Software, algorithm | Jupyter Notebook | https://jupyter.org/ | RRID:SCR_018315 | - |
| Software, algorithm | FSL | http://www.fmrib.ox.ac.uk/fsl/ | RRID:SCR_002823 | - |
| Software, algorithm | Tracker-MRI: Experiment control software | Netherlands Institute for Neuroscience | RRID:SCR_021816 | In-house developed, available via: https://github.com/VisionandCognition/Tracker-MRI |
| Other | NMT v1.3 | NIH, AFNI https://afni.nimh.nih.gov/pub/dist/doc/htmldoc/nonhuman/macaque/template_nmtv1.html#nmt-v1-3 | n/a | Macaque Brain Template and Atlas |

## Subject details

Four male macaques (*M. mulatta;* 7–12 kg, 5–8 years old) participated in this study. Animal care and experimental procedures were in accordance with the ILAR's Guide for the Care and Use of Laboratory Animals, the European legislation (Directive 2010/63/EU), and approved by the Institutional Animal Care and Use Committee of the Royal Netherlands Academy of Arts and Sciences and the Central Authority for Scientific Procedures on Animals (CCD) in the Netherlands (license numbers AVD8010020173789 and AVD8010020171046). The animals were socially housed in an enriched specialized primate facility with natural daylight, controlled temperature and humidity, and fed with standard primate chow, supplemented with raisins, fresh fruits, and vegetables. Their access to fluid was controlled, according to a carefully designed regime for fluid uptake. During weekdays, the animals received diluted fruit juice in the experimental setup. We ensured that the animals drank sufficient fluid in the setup and received extra fluid after experimental sessions if needed. On the weekends, animals received at least 700 ml of water in the home cage. The animals were regularly checked by veterinary staff and animal caretakers, and their weight and general appearance were recorded in an electronic logbook on a daily basis during fluid-control periods.

## Surgical procedures

Two animals (M1 and M2) participated in the MRI experiments and were implanted with an MRI-compatible plastic (PEEK) head-post, fixed to the skull with ceramic bone screws and acrylic (*Papageorgiou et al., 2014*; *Vanduffel et al., 2001*). Anesthetics, analgesics, and monitoring procedures were similar to previous surgical procedures in our laboratory and are described in detail elsewhere (*Klink et al., 2017*; *Poort et al., 2012*; *Supèr and Roelfsema, 2005*). Two other animals (M3 and M4) participated in the electrophysiology experiments. They were implanted with a custom 3D-printed titanium head-post that was designed in-house based on a CT scan of the skull, aligned to a T1-weighted anatomical MRI scan of the brain (*Chen et al., 2020*; *Chen et al., 2017*). The titanium head-post was attached to the skull with titanium bone screws, and the skin was closed around the implant without the use of any acrylic. In a second surgery, each animal was additionally implanted with a total of 1024 electrodes in 16 Utah electrode arrays (Blackrock Microsystems) in their visual cortices (14 arrays in V1, 2 arrays in V4; *Figure 1B*). Each array contained an 8-by-8 grid of 64 iridium oxide electrodes with a length of 1.5 mm spaced at a distance of 400 µm from each other. Pre-implantation electrode impedances ranged from 6 to 12 kΩ. A custom-designed 1024-channel pedestal was attached to the skull with titanium bone screws, and the skin was closed around it. More details on the surgical procedures have been published elsewhere (*Chen et al., 2020*; *Chen et al., 2017*).

## Visual stimuli and procedures

In the MRI experiment, animals were head-fixed, sitting in the sphinx position (*Papageorgiou et al., 2014*; *Vanduffel et al., 2001*), and viewing a 32″ screen (1920 × 1080 pixels, 100 Hz) (Cambridge Research Systems) at the end of the bore, 130 cm away. pRFs were measured using conventional moving bar stimuli that traversed the screen in eight different directions behind a large virtual circular aperture (*Figure 1*). The borders of this virtual aperture were invisible because both the foreground and background had the same gray level (22.3 cd/m²) (*Dumoulin and Wandell, 2008*). In the MRI experiments, the bar sweep spanned 16° (diameter) in 20 steps (*Dumoulin and Wandell, 2008*). The moving bars were 2° wide and contained a checkerboard pattern (100%  contrast; 0.5° checkers; luminance of white checkers: 106.8 cd/m²; luminance of black checkers: 0.2 cd/m²) that moved parallel to the bar's orientation. Each bar position was on the screen for 2.5 s (1 TR), making one full bar sweep last 50 s. Bar sweep series (all directions presented once) were preceded and followed by 37.5 s (15 TRs) of neutral gray background. Each horizontal or vertical bar sweep was followed by a neutral gray background period of 25 s. The order of the bar sweep directions was 270°, 315°, 180°, 225°, 90°, 135°, 0°, 45° on most runs, but for one animal we inverted the directions to 90°, 135°, 0°, 45°, 270°, 315°, 180°, 225° on some runs to compensate for the animal's tendency to fall asleep near the end of runs. We included data from 8 scanning sessions for monkey M1 (34 runs, 268 bar sweeps) and 10 sessions for monkey M2 (66 runs, 406 bar sweeps). During stimulus presentation, the animals received fluid rewards (Crist Instruments, Hagerstown, MD) for maintaining fixation within a circular fixation window with a diameter of 2°, centered on a 0.15° red fixation dot, surrounded by a 0.75° (diameter) aperture of neutral gray background color. In the electrophysiology experiments, the stimulus and task were very similar, but bar sweeps now spanned 28°, which was possible because the animals were closer to the monitor (luminance values of this monitor were black: 0 cd/m²; white: 92.1 cd/m², neutral gray: 14.8 cd/m²). Bars traveled along this path in 30 steps of 500 ms, and the neutral gray luminance intervals were reduced to 2.5 s due to the much faster neuronal responses (compared to the BOLD signal). The fixation window in the electrophysiology experiments was slightly smaller with a diameter of 1.5°. In the MRI experiment, eye position and pupil diameter were tracked with an MRI-compatible infrared eye-tracking system at 120 Hz (ISCAN ETL-200). Hand positions were also monitored using fiber optic amplifiers (Omron E3X-NH) and optic fibers. To reduce body movement-related imaging artifacts, the animals were trained to maintain their hands inside a response box by making reward delivery contingent on both eye and hand position. In the electrophysiology experiments, animals were head-fixed in a conventional vertical primate chair and viewed a 21″ CRT monitor (1024 × 768, 85 Hz) at a distance of 64 cm while their eye position and pupil diameters were tracked at 230 Hz using an infrared eye tracker (TREC ET-49B, Thomas Recording GmbH).

## MRI acquisition

MRI was performed in a standard Philips Ingenia 3.0T horizontal bore full-body scanner (Spinoza Center for Neuroimaging, Amsterdam, the Netherlands). We used a custom-built eight-channel phased array receive coil system (*Ekstrom et al., 2008*; *Kolster et al., 2009*) (KU Leuven) and the scanner's full-body transmit coil. Functional images were obtained using a gradient-echo T2* echo-planar sequence (44 horizontal slices, in-plane 72 × 68 matrix, TR = 2500 ms, TE = 20 ms, flip angle = 77.2°, 1.25 × 1.25 × 1.25 mm isotropic voxels, SENSE-factor of 2 in the AP direction, and phase-encoding in the AP direction).

## fMRI preprocessing

All fMRI data were preprocessed with a custom-written Nipype pipeline that we have made available online (RRID:021813; https://github.com/visionandcognition/NHP-BIDS). In short, MRI scans were exported from the scanner as DICOM images and converted to NIFTI files with the dcm2niix tool (*Li et al., 2016*). The volumes were then reoriented to correct for the animal being in the sphinx position and resampled to 1 mm$^3$ isotropic voxels. The resulting images were realigned using a nonrigid slice-by-slice registration algorithm based on AFNI tools (*Cox, 1996*) followed by an FSL-based motion correction procedure MCFLIRT (*Jenkinson et al., 2002*). Functional volumes were linearly aligned to the individual high-resolution anatomical volumes, which were in turn nonlinearly registered to the NMT standard space (*Seidlitz et al., 2018*). Preprocessed data were further processed with a combination of custom-written MATLAB (MathWorks, Natick, MA) and shell scripts (https://gin.g-node.org/ChrisKlink/NHP-PRF). BOLD time courses for each voxel were normalized to percentage signal change and averaged across runs (or parts of runs) for which fixation was maintained at 80% of the time or more. We averaged odd and even runs separately to allow for a cross-validation approach in the evaluation of the pRF model fits. Anatomical ROIs were defined based on a probabilistic atlas (*Reveley et al., 2017*; *Seidlitz et al., 2018*) and refined using individual retinotopic maps.

Post-fit comparisons across pRF models, HRFs, and ROIs were performed in MATLAB based on the volumetric results. For visualization of the fMRI data, volumetric results were also projected to the individual cortical surfaces. To create these surfaces, we averaged multiple anatomical scans (T1-weighted, 3D-FFE, TE = 6 ms, TR = 13 ms, TI = 900 ms, flip angle = 8°, 100 horizontal slices, in-plane 224 × 224 matrix, 0.6 × 0.6 × 0.6 mm isotropic voxels, and phase-encoding in the AP direction) and processed the result with customized tools based on FreeSurfer (*Fischl, 2012*) and Pycortex (*Gao et al., 2015*) that were adjusted to handle our NHP data. These tools and their documentation can be found at https://github.com/VisionandCognition/NHP-Freesurfer; (*Klink, 2021*; copy archived at swh:1:rev:8d4b89337b865fb194e196cf1b2af4967e14d607) and https://github.com/VisionandCognition/NHP-pycortex, respectively (*Messinger et al., 2021*).

## Electrophysiology acquisition

Neuronal activity was acquired from 1024 channels simultaneously at a 30 kHz sampling rate. The 1024-channel pedestal was connected to eight 128-channel CerePlex M head-stages through an electronic interface board. Each head-stage processed signals from two 64-channel electrode arrays with a 0.3–7500 Hz analog filter at unity gain (i.e., no amplification). After analog-to-digital conversion, the signal from each head-stage was sent to a 128-channel Digital Hub (Blackrock Microsystems) where it was converted into an optical output signal and sent to a 128-channel Neural Signal Processor (NSP, Blackrock Microsystems) for storage and further processing. The eight NSPs were controlled with eight simultaneously running instances of the Blackrock Central Software Suite (Blackrock Microsystems) distributed over two computers (four instances each) (*Chen et al., 2020*).

## Electrophysiology data preprocessing

The neuronal signal that was acquired using different software instances was temporally aligned using common TTL pulses sent by the stimulus computer. The data were then separated in (1) envelope MUA and (2) broadband LFP. MUA represents the spiking activity of a local population of neurons around the electrode (*Cohen and Maunsell, 2009*; *Palmer et al., 2007*; *Supèr and Roelfsema, 2005*). To extract MUA, we amplified the raw neuronal signal, band-pass filtered it between 500 Hz and 9 kHz, full-wave rectified it, and applied a low-pass filter of 200 Hz. The resulting time series were downsampled to 1 kHz. We subtracted the baseline MUA activity in a 1000 ms prestimulus time

window. Baseline-corrected MUA responses were then averaged, first across runs and then within a 50–500 ms time window for each stimulus position. The broadband LFP signal was generated by low-pass filtering the raw signal at 150 Hz and downsampling it to 500 Hz. The LFP signal was further processed with a multi-taper method using the Chronux toolbox (*Bokil et al., 2010*). Power spectra were calculated in a 500 ms moving window (step size 50 ms), using a time bandwidth product of five and nine tapers. LFP power was averaged within five distinct frequency bands: 4–8 Hz (theta), 8–16 Hz (alpha), 16–30 Hz (beta), 30–60 Hz (low gamma), and 60–120 Hz (high gamma). Baseline power in a 1000 ms prestimulus period was subtracted and the power for each recording site was averaged across runs, within a 50–500 ms time window during each stimulus position.

## pRF models and fitting procedure

We fit four pRF models to all the data (voxels and electrode channels) using a customized version of the analyzePRF toolbox (*Kay et al., 2013*) for MATLAB. In the fitting procedure, the stimuli were spatially downsampled to a resolution of 10 pixels per dva and converted to 'effective stimuli,' consisting of binary representations that encode stimulus position. Response predictions were calculated as the product of the effective stimulus and the pRF shape (*Equation 1*). This method also yields a prediction for pRFs that are only partially stimulated, and the best-fitting pRF can have a center location outside of the stimulus aperture. Because pRF estimates are generally more reliable for pRF that are strongly driven by a visual stimulus, we include indications of the directly stimulated visual field in our figures. The four pRF models differed in the pRF shape. Linear models (*Equations 1–3*) describe a single isotropic 2D Gaussian-shaped pRF and assume linear spatial summation across the visual field (*Dumoulin and Wandell, 2008*). We implemented two linear model versions. For the first model, responses were constrained to be positively related to the visual stimuli (P-LIN). A second version lacked this constraint and also allowed negative responses, that is, stimulus-driven activity reductions (U-LIN). Negative BOLD responses have been demonstrated in some brain areas (*Shmuel et al., 2006*; *Szinte and Knapen, 2020*). The nonlinear spatial summation model (*Kay et al., 2013*) expands the linear model by capturing nonlinear summation of signals across the visual field. It has previously been shown that the value of the exponent is generally smaller than 1 in human visual cortex, indicating subadditive spatial summation or CSS (*Kay et al., 2013*). This model is therefore generally referred to as the CSS model. Because the pRF size and static nonlinearity interact in the nonlinear model, the pRF size is defined as the standard deviation of the predicted Gaussian response profile to a point stimulus for all models (Equation 3; *Kay et al., 2013*). The mathematical descriptions of the linear and nonlinear pRF models are in *Equations 1–3*, where $Resp_{pred}$ indicates the predicted response, $g$ is a gain factor to scale the response, $S(x,y)$ is the effective stimulus, $G(x,y)$ is the Gaussian pRF profile, and $n$ is the exponent that determines the static spatial nonlinearity. For the P-LIN model, the gain $g$ was constrained to positive values, while for the U-LIN model, gain values could be negative as well. Negative gain factors imply stimulus-induced reductions of activity. In both linear models, the exponent $n$ was fixed to be 1. In the definition of the Gaussian, $(x_0, y_0)$ defines the center and $\sigma$ the standard deviation of the pRF.

Linear and nonlinear spatial summation pRF models:

$$\text{Resp}_{\text{pred}} = g \cdot \left[ \sum_{x,y} S\left(x,y\right) G\left(x,y\right) \right]^n, \quad \begin{array}{l} \text{P-LIN, U-LIN}: n = 1 \\ \text{P-LIN, CSS}: g > 0 \end{array} \tag{1}$$

$$G\left(x,y\right) = e^{\frac{-(x-x_0)^2 + (y-y_0)^2}{2\sigma^2}} \tag{2}$$

$$pRF_{\text{size}} = \frac{\sigma}{\sqrt{n}} \tag{3}$$

The DoG model uses a standard linear Gaussian ($G_1$ in *Equation 4*) to describe the excitatory center of a pRF and subtracts a second Gaussian profile ($G_2$) to model an inhibitory surround component (*Zuiderbaan et al., 2012*). This second Gaussian is by definition broader than the one describing the center. The size $\sigma_2$ and amplitude $a$ of the surround Gaussian are additional parameters (*Equations 4–6*). DoG pRF model:

$$Resp_{\text{pred}} = g \cdot \left[ \sum_{x,y} S\left(x,y\right) \left( G_1\left(x,y\right) - a \cdot G_2\left(x,y\right) \right) \right]^n \tag{4}$$

$$G_1\left(x,y\right) = e^{\frac{-(x-x_0)^2 + (y-y_0)^2}{2(\sigma_1)^2}}, G_2\left(x,y\right) = e^{\frac{-(x-x_0)^2 + (y-y_0)^2}{2(\sigma_2)^2}} ; \sigma_2 > \sigma_1 \tag{5}$$

The models assume a near-instantaneous link between the stimulus and the response dynamics, which holds for the electrophysiological signals. For these signals, the stimulus changes position every 500 ms, much slower than the onset or decay of activity. The BOLD response is much slower than the speed with which the stimulus traverses the screen in the fMRI experiments (2500 ms per position). We therefore convolved the predicted response with an HRF at a resolution of 1.25 s per sample (twice the acquisition rate of TR = 2.5 s). We used both a canonical human HRF and a standard monkey HRF that we derived from separate scanning sessions (*Figure 1—figure supplement 1*). In short, we presented the animals with brief (0.1 s) full-contrast and full-screen checkerboard stimuli. We then used FMRIB's Linear Optimal Basis Sets (FLOBS) (*Woolrich et al., 2004*) toolkit from the FSL software package to estimate the relative contributions of a set of basis functions for those voxels in the primary visual cortex that were activated by the stimulus. We then calculated a single weighted average HRF function based on these basis functions and used it as the standard monkey HRF. The monkey HRF was narrower than the canonical human HRF. It had a faster time-to-peak (4.2 s vs. 4.8 s) and peak-to-fall time (6.2 s vs. 12.6 s). The fits with the monkey HRF were slightly better, especially in the lower visual areas (*Figure 1—figure supplement 1*). However, size and location estimates were highly similar for the two HRFs for all models, and we only report results from model fits based on the monkey HRF.

Model fitting was performed on a cluster computer (LISA, SURFsara) using nonlinear optimization (MATLAB Optimization Toolbox). The accuracy of the different model fits was quantified as the cross-validated percentage of variance ($R^2$) explained of the BOLD response (*Equation 6*: *DATA*) by the model prediction (*Equation 6*: *MODEL*). For cross-validation, we divided the data into two nonoverlapping sets (odd and even runs) and tested the prediction of a model that was fit one data set against the time series of the other data set and vice versa. This yielded two $R^2$ values per voxel or electrode that were averaged. The cross-validated comparison of model performance is valid for models with different numbers of parameters and prevents overfitting. Fit results are available as voxel-based maps warped to the NMT template space on Neurovault.org (*Fox et al., 2021*; *Gorgolewski et al., 2015*) at https://identifiers.org/neurovault.collection:8082.

Model accuracy:

$$pRF_{\text{center-size}} = \sigma_1, pRF_{\text{surr-size}} = \sigma_2 \tag{6}$$

## Comparison of pRF (and cRF) estimates

After fitting the pRF models to all voxels and recording sites, we compared the pRF estimates both within and across recording modalities. We pooled voxels and recording sites across subjects and used nonparametric statistical tests (Wilcoxon signed-rank, Wilcoxon rank-sum, or Kruskal–Wallis) with post-hoc Tukey's HSD tests to correct for multiple comparisons. For the fMRI data, we compared $R^2$ across models and HRFs. We constructed retinotopic maps using the best pRF model and HRF and investigated the relationship between pRF eccentricity and size for a subset of ROIs with good model fits ($R^2 > 5\%$). For the electrophysiological data, we compared model accuracy and pRF estimates across MUA and LFP components to unravel to which extent retinotopic information is available in the different neuronal signals. We also compared the pRF estimates to a more conventional RF mapping technique for MUA based on responses to thin, moving bar stimuli (cRF). For recording sites with an SNR larger than 3 (i.e., visual responses that were more than three times larger than the standard deviation of the spontaneous activity), we fitted a Gaussian to the averaged MUA traces and determined the onset and offset of the visual response as the mean of this Gaussian plus or minus its standard deviation (SD). Horizontal and vertical RF boundaries were then derived from the onset and offset times for stimuli moving in opposite directions (*Supèr and Roelfsema, 2005*). RF centers were defined as the midpoint between the horizontal and vertical borders. For comparison with the pRFs, we calculated RF sizes as half the diagonal of the rectangular area between the horizontal and vertical cRF borders ($cRF_{sz} = \sqrt{(width^2 + height^2)}/2$). This measure approximates the RF radius based on the SD of a Gaussian response profile and can be directly compared to the sigma estimated by pRF models (*Figure 7*). It is smaller than cRF sizes typically reported in electrophysiological studies because neurons are activated by stimuli farther than 1 SD from their RF center (our lab usually defines cRF diameter as 3.3 * SD).

To compare pRFs based on fMRI-BOLD and electrophysiology, we combined data from individual animals to create one pool of BOLD-based voxel pRFs and six pools of electrophysiology-based electrode pRFs (MUA, and the different frequency bands of the LFP) for V1 and V4 data. We compared the relationship between RF eccentricity and size with a set of LMMs. We first tested for a correlation between eccentricity and size for the different signal types. Signal types with a significant correlation were subsequently tested together in a single LMM to determine whether the eccentricity-size relationship differed between signal types (interaction SIGNAL × ECC). Finally, we compared electrophysiological signals with the MRI results. For this analysis, we only selected V1 and V4 voxels with pRFs falling within the eccentricity range of the electrode arrays and voxels and electrodes with a fit accuracy above a predetermined threshold (fMRI threshold: $R^2 > 5\%$, electrophysiology threshold: $R^2 > 50\%$). To test the robustness of the results for this analysis, we repeated it with different combinations of criteria for initial voxel inclusion (i.e., including only voxels that roughly correspond to the location of the electrode arrays, or including all V1 and V4 voxels) and fit quality (fMRI threshold: $R^2 > 5\%$ or 10%, electrophysiology threshold: $R^2 > 25\%$ or 50%; *Figure 12—figure supplement 1*).

## Acknowledgements

We thank Jonathan Williford for his contributions to the fMRI preprocessing pipeline; Pieter Buur, Wietske van der Zwaag, Diederick Stoffers, and the Laboratory of Neuro- and Psychophysiology of KU Leuven for technical assistance in setting up the nonhuman primate MR infrastructure; Kor Brandsma, Anneke Ditewig, and Lex Beekman for animal care and biotechnical assistance; Feng Wang for help with electrophysiology data collection; Chris van der Togt for help with data management; and Tomas Knapen and Serge Dumoulin for fruitful discussion and comments on an earlier version of the manuscript. This work was supported by NWO (Crossover Program 17619 'INTENSE'; STW-Perspectief P15-42 'NESTOR'; VENI 451.13.023), the European Union FP7 (ERC 339490 'Cortic_al_gorithms'), the Human Brain Project (agreements 720270 and 785907, 'Human Brain Project SGA1 and SGA2'), and the Friends Foundation of the Netherlands Institute for Neuroscience.

## Additional information

### Funding

| Funder | Grant reference number | Author |
|---|---|---|
| Nederlandse Organisatie voor Wetenschappelijk Onderzoek | VENI 451.13.023 | P Christiaan Klink |
| Nederlandse Organisatie voor Wetenschappelijk Onderzoek | STW-Perspectief P15-42 "NESTOR" | Xing Chen Pieter Roelfsema |
| FP7 Ideas: European Research Council | ERC 339490 "Cortic_al_gorithms" | Pieter R Roelfsema |
| Human Brain Project | Agreements 720270 and 785907 "Human Brain Project SGA1 and SGA2" | Pieter R Roelfsema Wim Vanduffel |
| Nederlandse Organisatie voor Wetenschappelijk Onderzoek | Crossover Program 17619 "INTENSE" | Pieter R Roelfsema P Christiaan Klink |

The funders had no role in study design, data collection and interpretation, or the decision to submit the work for publication.

### Author contributions

P Christiaan Klink, Conceptualization, Data curation, Formal analysis, Funding acquisition, Investigation, Methodology, Project administration, Resources, Software, Visualization, Writing – original draft, Writing – review and editing; Xing Chen, Formal analysis, Funding acquisition, Investigation, Writing

– review and editing; Wim Vanduffel, Methodology, Writing – review and editing; Pieter R Roelfsema, Conceptualization, Funding acquisition, Resources, Supervision, Writing – review and editing

### Author ORCIDs
P Christiaan Klink  http://orcid.org/0000-0002-6784-7842
Xing Chen  http://orcid.org/0000-0002-3589-1750
Pieter R Roelfsema  http://orcid.org/0000-0002-1625-0034

### Ethics

Animal care and experimental procedures were in accordance with the ILAR's Guide for the Care and Use of Laboratory Animals, the European legislation (Directive 2010/63/EU) and approved by the institutional animal care and use committee of the Royal Netherlands Academy of Arts and Sciences and the Central Authority for Scientific Procedures on Animals (CCD) in the Netherlands (License numbers AVD8010020173789 and AVD8010020171046).

### Decision letter and Author response

Decision letter https://doi.org/10.7554/eLife.67304.sa1
Author response https://doi.org/10.7554/eLife.67304.sa2

---

## Additional files

### Supplementary files

• Supplementary file 1. Table with region of interest (ROI) abbreviations. List of ROI abbreviations, color coded by where in the brain they are located.

• Transparent reporting form

### Data availability

All data and code are available on GIN: https://doi.gin.g-node.org/10.12751/g-node.2j01af. Unthresholded fMRI model fitting results are available on Neurovault: https://identifiers.org/neurovault.collection:8082.

The following dataset was generated:

| Author(s) | Year | Dataset title | Dataset URL | Database and Identifier |
|---|---|---|---|---|
| Klink PC, Chen X, Vanduffel W, Roelfsema PR | 2021 | Visual population receptive fields - macaque fMRI | https://identifiers.org/neurovault.collection:8082 | NeuroVault, 8082 |
| Klink PC, Chen X, Vanduffel W, Roelfsema PR | 2021 | Population receptive fields in non-human primates from whole-brain fMRI and large-scale neurophysiology in visual cortex. | https://doi.org/10.12751/g-node.2j01af | GIN, 10.12751/g-node.2j01af |

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
