## [Editor Report]

This study is a detailed, systematic comparison of visually evoked population receptive fields (pRFs) measured non-invasively with MRI and invasively with electrophysiology in the same primate species. The authors show that MRI pRFs provide a good estimate of receptive fields based on multi-unit spiking activity in early visual areas. These results make an important contribution to our understanding of human imaging data in research and in the clinic.

---

## [Decision Letter]

**Decision letter after peer review:**

Thank you for submitting your article "A comparison of population receptive fields from fMRI and large-scale neurophysiology recordings in non-human primates" for consideration by *eLife*. Your article has been reviewed by 3 peer reviewers, one of whom is a member of our Board of Reviewing Editors, and the evaluation has been overseen by Timothy Behrens as the Senior Editor. The following individual involved in review of your submission has agreed to reveal their identity: Holly Bridge (Reviewer #2).

Essential revisions:

1) Framing of the study:

– Title and Abstract should differentiate more clearly that the direct comparison between neuronal activity and MRI pRFs was done only for V1 and V4 only. The wider MRI dataset was used for assessment of pRF models. The abstract felt somewhat misleading because it suggests there is neurophysiological data from areas other than V1 and V4. It should be made clear from the outset that neurophysiological data are only available in these areas. The statement 'We found that pRFs derived from BOLD-fMRI were most similar to MUA-pRFs in areas V1 and V4…..' makes it sounds like they were compared in other areas, but were most similar in those ones.

– In terms of improvement in the manuscript, we had difficulty determining the precise hypothesis or aim of the manuscript because there seem to be 3 different ideas (1) the relationship of pRF with eccentricity across many cortical and subcortical areas (2) 4 different models for fitting the pRF model and (3) the comparison of BOLD pRF with MUA and LFP in areas V1 and V4. It would therefore be helpful to understand better from the manuscript what the authors intend to be the major question.

2) In some cases, pRFs from higher areas in the frontal lobe etc appear to be coming from just one animal and the number of voxels meeting the R2 criterion is very low. We would suggest that analyses are limited to regions where pRF can be fit in both animals (and with a threshold for the minimum number of voxels). The entry criteria used should be reported in the Results.

3) Figure 3. LGN and Pulvinar:

a) The legend states that both areas have retinotopic maps of the contralateral visual field. We would definitely agree with this assertion for the LGN, but this does not appear to be the case for the pulvinar. Is this quantified somewhere?

b) Figures 3A and 3B suggest that M1 and M2 had inverted retinotopic maps in the LGN in one hemisphere between the two animals. M1 shows upper visual field (yellow) dorsally and M2 shows upper visual field ventrally (yellow; sagittal view). The LGN retinotopic maps do not look consistent between the two monkeys, which suggests that they might not as reliable as stated in the results.

4) Figure 5 really needs some data points rather than just the fits. We realise all the voxels are shown in the supplementary materials, but showing binned data with errorbars would be very helpful. It would also be more transparent, given the considerable differences in voxel numbers (mentioned above), to have the saturation of the data and fits reflect the voxel counts. That would also remove the issue with the lines intersecting the y-axis at or below zero as this is difficult to explain.

5) For the neurophysiology and MRI comparisons, we wonder whether the authors could provide a more direct comparison of the data considering only the pRF data from the regions contributing to the neurophysiological recordings.

6) The discussion of the pulvinar seems out of place in this manuscript as it does not seem to be the point of the paper and there is little analysis of the data from this area (noted above that representation does not appear to be consistently contralateral).

7) Some critical details are missing in the methods with regards to the assessed visual field and the stimulus used:

a) In the results and in methods, the covered visual field is described as 16 degrees (MRI) and 28 degrees (neurophysiology). Since this critically limits the RF size-eccentricity relationship that can be assessed, the authors should clearly state whether this is the diameter or radius of the visual field and whether (and how) horizontal and vertical dimensions of the available visual field differed. We assume that the stimulated visual field is only up to 8 degrees eccentricity for the MRI data – this needs to be clearly stated and in every data figure with actual data points and fits/extrapolations distinguished.

b) If partially stimulated receptive fields are extrapolated, then these need also to be indicated where applicable in the results, their generation explained in the methods, and the implications discussed, as the precise shape of these depends on the assumptions made.

c) Also, the results (line 130) refer to a "large circular aperture" behind which the stimulus was shown and the results refer here to Figure 1. The stimulus in Figure 1 shows no such aperture. Details of this (size, position, contrast) should be provided.

d) Mean luminance of the monitors used and the stimulus contrast should be provided for both set ups in the Methods.

8) Following on from 7), given the methods descriptions, it is not clear how some foveal or more eccentric RFs were assessed.

a) The results state clearly that the animals fixated the screen centrally. If the visual field refers to the diameter, this means that RFs more eccentric than 8 degrees (MRI) and 14 degrees (neurophysiology) could not be assessed.

This potentially affects the results in Figures 5 and 6B (M4). In Figure 5, the authors should distinguish between the range based on available data and extrapolation for each area.

b) Conversely, according to the methods around the fixation dot there was mask of mean background colour (aperture of 0.75 degrees – we assume radius?). But in Figure 6A, B and C, there are RF centres within 0.5 degrees of the fixation point. The authors should add an explanation to the methods how they assessed the RFs in these position, if they did not fully stimulate this part of the visual field.

9) Figure 7 suggests a systematic overestimation of pRF size over cRF? It would be good to compare both measures quantitatively to the neurophysiological literature, so the readers can understand what "the different definitions of RF size" refer to (p11, l246-247; p13, l281-283). The comparison as is in results and discussion remains vague as to RF definition and systematic differences across neurophysiology – MRI literature in estimated RF sizes.

10) Could the authors provide measures about the mean, range and variance in eye position for the MRI and neurophysiology experiments, as this would have potentially affected the size estimates of the smaller RFs. It would be good to discuss this issue.

11) Figure 6B suggests that the neurophysiology data for V4 samples two separate parts of the visual field in the two monkeys.

(a) Could the authors state in the paper whether there is in any of their analyses a significant difference in the data from the two monkeys and whether this could have affected the comparison with the MRI?

(b) Figure 6B suggests that in M4, by far the more eccentric visual field positions were sampled. However, Figure 7B suggests that most of the measured RF sizes are the same or smaller than in M3. Is this correct? And how does this relate to the literature on V4 RF sizes?

*Reviewer #1:*

This study provides a systematic assessment of visual population receptive fields (pRF) obtained with functional magnetic resonance imaging (MRI) in primates.

Critically, pRFs from early visual areas V1 and V4 are compared to systematic neurophysiological measurements obtained in individuals from the same primate species (but not the same individual animals) across a range of visual field positions. This allows at a direct comparison to "ground truth".

Furthermore, the paper provides an assessment of four different models for generating receptive fields from imaging and neurophysiological data, including linear and non-linear models, and investigates multi-unit (MUA) and local field potential (LFP) data. The authors show the greatest correspondence of pRFs between MUA and fMRI data, but the reasons for this could be more fully explored.

The paper could situate the results more clearly in the wealth of data we have about neurophysiological measures of receptive fields and retinotopy for the macaque visual system. The presented results could differentiate more clearly between data for which MRI and neurophysiological data are available (central visual field representations for areas V1 and V4), data for which extrapolation partially stimulated RFs was included and results that depend on MRI data only. Also, outside the early visual cortical areas the MRI retinotopic maps appear potentially less reliable and with this the pRF measurements.

These results are are equally important for research and clinical assessments of the neuronal basis of visual functional and perception.

*Reviewer #2:*

This paper by Klink and colleagues investigates the relationship between visual receptive fields defined with non-invasive magnetic resonance imaging (MRI) and neurophysiology in the non-human primate. Population receptive field mapping (pRF) is a non-invasive imaging technique designed to estimate the area of the visual field to which particular visual brain areas respond. Since these pRFs are derived from MRI data, they have provided a window into human visual brain function. In contrast, using invasive neurophysiology, it is possible to measure the receptive field from a group of nearby neurons (multi-unit activity; MUA) or the local field potential (LFP).

Data from four animals are presented, with pRF MRI data acquired from two animals and neurophysiological measurement of both MUA and LFP in the other two with some comparison between the measurements.

Strengths of the study:

1. The comprehensive analysis of the pRF mapping, which compares the results of four different models. The data are of very high quality due to the relatively high resolution of the imaging and the large quantity of data acquired. This allowed the identification of many areas in the occipital lobe, along with parietal, temporal and even some frontal regions. It was also possible to map pRFs in both LGN and pulvinar.

2. The finding that negative BOLD signals were due to two different sources – (i) being outside of the region that was visually stimulated and (ii) regions that are part of the default mode network – is an elegant finding and is consistent with previously published human data.

3. Recording both MUA and LFP power at different frequency bands provides a number of different measures of receptive field size that can be compared to the pRF data, and indeed the correspondence seems high.

4. Number of recording sites in V1 is good giving a good range of eccentricities over which to measure receptive fields.

Weaknesses of the study:

1. The authors are not clear about the main message of the paper. The title suggests that it is the comparison of the pRF and neurophysiological findings, but the neurophysiological recordings are (understandably) predominantly in V1, with some also in V4. This means that the majority of the pRF data cannot be compared to the neurophysiological recordings. Additionally, that the data were acquired in different animals is unfortunate as it would have been particularly valuable to measure the neurophysiological data from cortical tissue where pRFs have been measured non-invasively.

2. Four different methods for measuring pRFs were used, but the authors do not draw strong conclusions about which model may be the most successful to use in the discussion. Having this information would be very valuable for other researchers.

3. The discussion could be more focussed and does not really provide the reader with an answer to the question of how well the pRF compares to the neurophysiological data. Much of the data on pRF size and eccentricity has been published many times in the human and there is already some data in the non-human primate, so it is important to emphasise the novel findings of the current data. I would expect some more in-depth explanation of the MUA and LFP relationship with pRF and why MUA appears better.

4. Since data are from 2 animals, it would be helpful to see which animal each datapoint comes from to ensure that there are no major biases due to the contribution of each animal.

While the conclusions are supported to a certain extent, the key point that is critical to highlight is that the neurophysiological data are predominantly from V1. It is too general to state that MRI-based pRF measurements reliably reflect receptive field properties of the primate brain.

*Reviewer #3:*

The authors present fMRI data from 2 macaques and neural electrophysiology data from 2 different macaques implanted with 16*64 electrodes covering the lower right visual field quadrant of V1, and V4. Stimuli consisted of a traveling bar that systematically stimulated the retina while awake animals fixated centrally. pRFs fit to BOLD signals evoked by these stimuli, were compared with pRFs fit to time courses from multi unit activity (MUA) as well LPF theta, α, β, low γ and high γ frequency power.

This is an impressive study that offers a first systematic and very thorough comparison of pRFs estimated from FMRI versus from LFP as well as MUA electrophysiology measures. It thereby contributes to validation and interpretation of an increasing body of human pRF mapping research. The analyses are thorough and conclusions drawn are well supported by the data in my view (though note that I am not an expert on electrophysiological recording methods). A particular strength of the approach taken, is that the authors have fit 4 different pRF models to all these data using a robust cross-validation approach, which allows them to capture and compare various types of deactivation and spatial summation dynamics, and make inferences about parallels and differences in spatial information processing across brain regions and neural activity signatures. Most crucially, results convincingly reveal accuracy of fMRI pRFs with electrophysiology-recordings of MUA and γ LFP, in terms of similar relationships between pRF size, eccentricity, and the best-fitting pRF model parameters (a 'compressive spatial summation' model). Another strength includes the investigation of wide-spread retinotopy with fMRI, confirming recent findings of retinotopic selectivity in brain regions not classically considered visual or even sensory, such as the default mode network, in humans.

In terms of weaknesses, I see no major issues, although there are some aspects of the work that could be clarified. First, it would be helpful if the authors could elaborate on how pRF size estimated from MUA and LPF measures depend on the size or sensitivity of the electrodes to various neural signals, and what this means for expected comparability of pRF parameter estimates across these measures and to MRI. Second, there were a few points I felt lacked some detail needed to understand the results: one intriguing finding, is that for some electrodes in V1, pRFs can be derived from α and β frequencies in LFP signal that differ from those in the γ frequency. Interestingly, there are negative or 'deactivating' pRFs measurable in α and β LFP signals, which are substantially shifted in position with respect to 'activating' pRFs derived from the γ frequency LFP at the same electrode. The authors speculate this reflects foveally-directed shifts due to attention or eye-movement, each of likely relevance to studies mapping near-foveal regions. However, I found the relevant plots collapsing across electrodes (Figure 10) quite hard to interpret, and wondered whether differences in pRF fit across the two data types (Supplementary Figure 7) may contribute to these shifts. Similarly, in another very interesting analysis the authors show that pRF sizes fit to the MUA response evoked by the mapping stimulus, correlates well with RF size estimates obtained with the 'classic' approach of recording the MUA response to a thin moving light bar. Data reveal a clear linear relationship in RF size estimates across measures, but an analysis of expected vs. observed differences in size and shape (i.e., the true identity line) that would be useful for assessing methodological biases is not provided.

So, barring some minor points of clarification, I believe this work offers a wealth of data and analyses that provide important validation of commonly used pRF mapping methods, and will without doubt spark many further investigations into population receptive field dynamics across the primate and human brain. This will be hugely facilitated by the free availability of data and code for this study, made accessible via a link in the paper.

---

## [Author Response]

Essential revisions:1) Framing of the study:– Title and Abstract should differentiate more clearly that the direct comparison between neuronal activity and MRI pRFs was done only for V1 and V4 only. The wider MRI dataset was used for assessment of pRF models. The abstract felt somewhat misleading because it suggests there is neurophysiological data from areas other than V1 and V4. It should be made clear from the outset that neurophysiological data are only available in these areas. The statement 'We found that pRFs derived from BOLD-fMRI were most similar to MUA-pRFs in areas V1 and V4…..' makes it sounds like they were compared in other areas, but were most similar in those ones.– In terms of improvement in the manuscript, we had difficulty determining the precise hypothesis or aim of the manuscript because there seem to be 3 different ideas (1) the relationship of pRF with eccentricity across many cortical and subcortical areas (2) 4 different models for fitting the pRF model and (3) the comparison of BOLD pRF with MUA and LFP in areas V1 and V4. It would therefore be helpful to understand better from the manuscript what the authors intend to be the major question.

Thank you for pointing this out. Our main objective was to compare BOLD-pRFs and neurophysiology pRFs. However, because the data-set is rich, we also explored it by fitting different pRF models and looking at brain areas beyond V1 and V4 in the fMRI results. While we feel that all these aspects of the data yield interesting insights, we agree that this broadening of the scope may have obscured the main question a bit.

We changed the title of the revised manuscript to ‘Population receptive fields in non-human primates from whole-brain fMRI and large-scale neurophysiology in visual cortex’ and more clearly differentiate the main question of the BOLD-ephys comparison from the complementary findings in the revised abstract. The title is succinct as prescribed by *eLife*, while also explicitly mentioning the scope of neurophysiological and neuroimaging recordings. The abstract now also clarifies that direct comparisons were made only for visual areas V1 and V4. It now also makes clear that this comparison was our main question, but that additional insights were derived from the broader scope of the fMRI data. This distinction is now also properly specified in the introduction.

2) In some cases, pRFs from higher areas in the frontal lobe etc appear to be coming from just one animal and the number of voxels meeting the R2 criterion is very low. We would suggest that analyses are limited to regions where pRF can be fit in both animals (and with a threshold for the minimum number of voxels). The entry criteria used should be reported in the Results.

This assertion is correct. Selecting data with a stricter criterion primarily affects Figure 5, which we have now changed to address this and other reviewer suggestions. As suggested by the reviewers, we now only show areas for which at least 20 voxels reach the R^2^ threshold in both animals and explicitly mention how many voxels meet this threshold in each individual. This inclusion criterion is also explicitly mentioned in the caption of the new Figure 5. For more details see point 4.

3) Figure 3. LGN and Pulvinar:a) The legend states that both areas have retinotopic maps of the contralateral visual field. We would definitely agree with this assertion for the LGN, but this does not appear to be the case for the pulvinar. Is this quantified somewhere?

We thank the reviewers for these observations and agree with them that the pulvinar results are not very consistent across animals. We have changed our phrasing in the legend to reflect this better. We also added a quantification of the proportion of voxels with significant contralateral pRFs (n_contra_/n_total_). The new text in the legend reads:

“The lateral geniculate nucleus (LGN, top rows) contained retinotopic maps of the contralateral visual field in both monkeys (M1: 38/38; M2: 73/80 voxels with contralateral pRFs). Retinotopic information was also present in the pulvinar (PULV, bottom rows), but its organization was much less structured, especially in M2 (M1: 23/32; M2: 61/131 voxels with contralateral pRFs).”

b) Figures 3A and 3B suggest that M1 and M2 had inverted retinotopic maps in the LGN in one hemisphere between the two animals. M1 shows upper visual field (yellow) dorsally and M2 shows upper visual field ventrally (yellow; sagittal view). The LGN retinotopic maps do not look consistent between the two monkeys, which suggests that they might not as reliable as stated in the results.

We did not perform a detailed analysis of the LGN and pulvinar maps, as we considered the resolution of our data to be unsuitable for such inferences. When we compared our results to previously published retinotopic maps of the LGN (e.g. Erwin et al., 1999; Schneider et al., 2004; DeSimone et al., 2015), we confirm the representation of the upper visual field in more anterior-ventral LGN, and of the lower visual field in posterior-dorsal LGN. This is less clear in M2 where fewer voxels could be fit, and while a similar pattern might be present in some sagittal slices, others appear to show the opposite pattern. Whereas pRF model fits revealed retinotopic information for the pulvinar as well, these did not really constitute a clear ‘map’. We added a brief discussion about the reliability of the subcortical results to the Discussion (p19, ln 387-390):

“Retinotopic maps in the LGN were roughly in line with previously published retinotopic organization of the macaque LGN (Erwin et al., 1999), but they comprised few voxels and the variability across animals prohibited detailed inferences. In the pulvinar, the organization of the retinotopic information was even more variable across animals.”

4) Figure 5 really needs some data points rather than just the fits. We realise all the voxels are shown in the supplementary materials, but showing binned data with errorbars would be very helpful. It would also be more transparent, given the considerable differences in voxel numbers (mentioned above), to have the saturation of the data and fits reflect the voxel counts. That would also remove the issue with the lines intersecting the y-axis at or below zero as this is difficult to explain.

We have re-plotted the results in a revised version of Figure 5 to address both this point and point 2 above. Visualizing the number of voxels with saturation in addition to indicating the visual area with different colors did not work very well, but we found a way to represent the requested information. In revised Figure 5, we (1) only show areas for which both animals contribute at least 20 voxels, (2) mention the voxel-counts per area per animal, (3) plot binned data with error bars, (4) show linear fits for multiple areas in different colors in the same panel without confidence intervals, (5) show linear fits with confidence intervals for separate areas in smaller panels on the side, and (6) indicate the extent of the visual stimulus to clarify which pRFs were only partially stimulated (those in the gray areas). We also added these binned data-points to the corresponding Figure 5—figure supplement 1, which we expanded with a version of Figure 5—figure supplement 2 in which we lowered the R^2^ threshold to 3%. Since we make all data and scripts available online, interested readers will be able to investigate the effect of different voxel selection criteria on the results.

5) For the neurophysiology and MRI comparisons, we wonder whether the authors could provide a more direct comparison of the data considering only the pRF data from the regions contributing to the neurophysiological recordings.

Ideally, one would like to compare the BOLD-pRFs and ephys-pRFs from identical brain locations. However, because the recordings come from different individuals, the comparison depends on the precision with which we can estimate the correspondence of voxels in M1 and M2 with the location of electrodes in M3 and M4. We chose to approach this comparison from a functional angle: as the ephys-pRFs were located in the lower right visual field, we only included V1 and V4 voxels for which the BOLD-pRFs were also located in the lower right visual field. We have now expanded the analysis with the subset of V1 and V4 voxels in the left hemisphere that roughly corresponded to the location of the electrodes implanted in M3 and M4. Data from these three different voxel inclusion criteria were furthermore analyzed with four different combinations of R^2^-based thresholds for the MRI and electrophysiology results. As a consequence, the analysis was run on 12 different combinations of inclusion criteria, yielding similar results. This approach is described on p18 ln 369-373 and an overview of the results is presented in the revised Figure 12—figure supplement 1:

“To further investigate the robustness of these results, we repeated the analysis, (1) with the inclusion of either all V1 and V4 voxels or a subset of the voxels in approximately the same anatomical location as the electrode arrays, and (2) by varying the R2-based inclusion criteria for electrodes and voxels (Figure 12—figure supplement 1). MUA-pRFs in V1 and V4 were generally similar to the BOLD-pRFs, although in V4 γ_low_ and γ_high_ also approximated the fMRI results in some of the comparisons.”

6) The discussion of the pulvinar seems out of place in this manuscript as it does not seem to be the point of the paper and there is little analysis of the data from this area (noted above that representation does not appear to be consistently contralateral).

We agree that the discussion of the (noisy) pulvinar could distract from the manuscript’s main points and have removed it. In addition, we extended the discussion of the subcortical results (as mentioned above in A3b).

7) Some critical details are missing in the methods with regards to the assessed visual field and the stimulus used:a) In the results and in methods, the covered visual field is described as 16 degrees (MRI) and 28 degrees (neurophysiology). Since this critically limits the RF size-eccentricity relationship that can be assessed, the authors should clearly state whether this is the diameter or radius of the visual field and whether (and how) horizontal and vertical dimensions of the available visual field differed. We assume that the stimulated visual field is only up to 8 degrees eccentricity for the MRI data – this needs to be clearly stated and in every data figure with actual data points and fits/extrapolations distinguished.

Yes, the assumption is correct and we have clarified the methods description accordingly (p25 ln 542-546). We have also added an indication of the extent of the visual stimulus in Figures 5,6 and Figure 5—figure supplements 1,2.

b) If partially stimulated receptive fields are extrapolated, then these need also to be indicated where applicable in the results, their generation explained in the methods, and the implications discussed, as the precise shape of these depends on the assumptions made.

The fitting procedure is indeed capable of estimating partially stimulated pRFs and all reported pRF estimates are fully data-driven and not extrapolated. Especially for larger pRFs just outside the visually stimulated region this works well. We have added an explanation of how the method deals with partially stimulated pRFs to the description of the methods (p28 ln 632-635).

“This method also yields a prediction for pRFs that are only partially stimulated, and the best fitting pRF can have a center location outside of the stimulus aperture. Because pRF estimates are generally more reliable for pRF that are strongly driven by a visual stimulus, we include indications of the directly stimulated visual field in our figures.”

c) Also, the results (line 130) refer to a "large circular aperture" behind which the stimulus was shown and the results refer here to Figure 1. The stimulus in Figure 1 shows no such aperture. Details of this (size, position, contrast) should be provided.

We have improved the description. The foreground and background of this aperture had the same color, which makes the aperture ‘virtual’. It clipped the bar stimuli, but its edges were not visible on the screen. We have now added a dashed line to the revised Figure 1A to indicate the virtual aperture, removed the mentioning of the aperture from the main text of the Results section to avoid confusion, and described the stimulus better in the Methods section:

“Population receptive fields were measured using conventional moving bar stimuli that traversed the screen in eight different directions behind a large virtual circular aperture (Figure 1). The borders of this virtual aperture were invisible, because both the foreground and background had the same gray level (22.3 cd/m2) (Dumoulin and Wandell, 2008).”

d) Mean luminance of the monitors used and the stimulus contrast should be provided for both set ups in the Methods.

We have added this information to the Methods.

8) Following on from 7), given the methods descriptions, it is not clear how some foveal or more eccentric RFs were assessed.a) The results state clearly that the animals fixated the screen centrally. If the visual field refers to the diameter, this means that RFs more eccentric than 8 degrees (MRI) and 14 degrees (neurophysiology) could not be assessed.This potentially affects the results in Figures 5 and 6B (M4). In Figure 5, the authors should distinguish between the range based on available data and extrapolation for each area.b) Conversely, according to the methods around the fixation dot there was mask of mean background colour (aperture of 0.75 degrees – we assume radius?). But in Figure 6A, B and C, there are RF centres within 0.5 degrees of the fixation point. The authors should add an explanation to the methods how they assessed the RFs in these position, if they did not fully stimulate this part of the visual field.

We have improved the methods description. The pRF fitting method is a forward modeling approach that predicts the responses to a visual stimulus for a hypothesized pRF (location and size) and minimizes the difference between the observed and predicted activity by changing the parameters of the hypothesized pRF. This means that (1) all estimated pRFs are data-driven, and (2) pRFs can be estimated even if their center is not stimulated. This holds for both the foveal (0.75 deg aperture was defined in diameter, which has been added to the Methods) and eccentric pRFs. Moreover, while we take the σ of the estimated pRF as a description of its size, the 2D Gaussians are larger than one σ, and a response occurs if a visual stimulus is farther from the center of the pRF. Figure 5 has been revised in light of other comments (see A4). In the new figure (as well as in other figures that deal with eccentricity), we added an indication of the extent of the visual stimulus to all figures that deal with eccentricity. We also added a few lines about cRF and pRF size definitions to the Methods section that will hopefully clarify this issue as well (p31 ln 718-722):

“This measure approximates the RF radius based on the SD of a Gaussian response profile, and can be directly compared to the σ estimated by pRF-models (Figure 7). It is smaller than cRF sizes typically reported in electrophysiological studies, because neurons are activated by stimuli farther than one SD from their RF center (our lab usually defines cRF diameter as 3.3*SD).”

9) Figure 7 suggests a systematic overestimation of pRF size over cRF? It would be good to compare both measures quantitatively to the neurophysiological literature, so the readers can understand what "the different definitions of RF size" refer to (p11, l246-247; p13, l281-283). The comparison as is in results and discussion remains vague as to RF definition and systematic differences across neurophysiology – MRI literature in estimated RF sizes.

This comment inspired us to recalculate the cRFs with a measure that is more comparable to the pRFs. In the previously submitted version of the manuscript, we took the position of a horizontally or vertically moving bar stimulus at visual field positions that included 95% of the area under a Gaussian fit to the MUA traces as the borders of the receptive field. Now we use the standard deviation of this Gaussian as a more direct analog to the pRF-size which is also quantified as 1 standard deviation. Interestingly, the cRFs calculated this way are smaller than pRFs estimated with the traditional P-LIN model (median difference 0.50 dva). This difference largely disappears when pRFs are calculated with the CSS model. Now, pRFs are slightly smaller than cRFs (median difference -0.13 dva). We have revised Figure 7 to show this direct comparison and changed the text where appropriate (p11 ln 247-251; p20 ln 439-443; p31 ln 714-722).

[p11 ln 247-251]”For this cRF method, we selected channels with a signal-to-noise ratio (SNR) larger than three (i.e., visual responses that were more than three times larger than the standard deviation of the spontaneous activity) and derived the RF borders from the onset and offset of the neuronal activity elicited by a smoothly moving light bar (see Materials and methods).”

[p20 ln 439-443]”For the MUA data, the CSS model estimated pRF sizes that were very similar to RF size estimates derived from conventional methods that use moving luminance bars, whereas the P-LIN model systematically returned larger pRF estimates. This difference might indicate that spatial compression indeed better captures the neuronal RF properties, at least in V1 and V4.”

[p31 ln 714-722]”Horizontal and vertical receptive field boundaries were then derived from the onset and offset times for stimuli moving in opposite directions (Supèr and Roelfsema, 2005). Receptive field centers were defined as the midpoint between the horizontal and vertical borders. For comparison with the pRFs, we calculated RF sizes as half the diagonal of the rectangular area between the horizontal and vertical cRF borders (cRFsz=(width2+height2)2). This measure approximates the RF radius based on the SD of a Gaussian response profile, and can be directly compared to the σ estimated by pRF-models (Figure 7). It is smaller than cRF sizes typically reported in electrophysiological studies, because neurons are activated by stimuli farther than one SD from their RF center (our lab usually defines cRF diameter as 3.3*SD).”

Various methods exist for the determination of RFs from spiking data, making comparisons across different labs somewhat difficult. The method we describe here is a procedure in use in our lab. To facilitate the comparison with the σ of pRFs we here use the σ value of a fitted Gaussian as measure for cRF size. However, we and other labs typically define the RF border at a multiple of this value (e.g., 1.65*σ) since neurons are activated by stimuli well beyond one σ from the center of their RF. In many of the papers we cite, RFs were manually marked based on subjective criteria derived from listening to neural activity (Gattass et al., 1987, 1981; Van Essen et al., 1994; Burkhalter and Van Essen, 1986; Rosa et al., 1988; Felleman and Van Essen, 1987; Newsome et al., 1986; Rosa et al., 2000; Gattass et al., 1988). Conversely, it is common in pRF studies to report the σ of the fitted Gaussian as measure of pRF size, but some studies also report pRF sizes as FWHM.

10) Could the authors provide measures about the mean, range and variance in eye position for the MRI and neurophysiology experiments, as this would have potentially affected the size estimates of the smaller RFs. It would be good to discuss this issue.

We used eye-trackers to monitor eye position and fixation performance determined reward delivery. We excluded epochs with insufficient fixation performance (<80%) from the subsequent model-fits and analysis. For all the recordings that were included in our pRF modeling pipeline, the monkeys had their gaze predominantly inside a fixation window that was slightly smaller in the electrophysiology recordings (0.75° radius) compared to the MRI experiments (1° radius) for practical reasons (vibrations from the scanner add a bit of noise to the eye position signal). We now determined the median eye position and the span of the IQR (in X and Y direction separately) for each animal across all sessions in both the MRI and electrophysiology experiments. The median eye positions for all animals was within 0.2° from the center of fixation, while the maximum IQR we observed was 0.6° for M4 in the electrophysiology recordings. Importantly, the IQR for eye position were similar for MRI and electrophysiology (MRI M1: IQRx = 0.2°, IQRy = 0.3°; M2: IQRx = 0.4°, IQRy = 0.5°; Electrophysiology M3: IQRx = 0.2°, IQRy = 0.4°; M4: IQRx = 0.4°, IQRy = 0.6°).

Brief and occasional eye-movements outside the fixation window and small residual eye-movements within the fixation window are unlikely to substantially affect the slow BOLD signal and BOLD-pRF estimates. Furthermore, they are likely to average out over stimulus set repetitions. Small residual eye-movements could cause a slight over-estimation of pRF-size. Moreover, the comparison of pRF size estimates across the various electrophysiology signals are unlikely to be influenced by eye movements, because the signals on which they are based were recorded at the same time (i.e., during the same eye-position variations).

Combining all of the above considerations, we consider it unlikely that eye movements play a significant role in our findings. In the discussion of the comparison of pRFs across signal types, we now briefly mention the potential effect of eye movements (p20, ln 427-432):

“Eye-movements can have an effect on the pRF size estimates and it is therefore important to note that (1) we only included data from recordings where the animals maintained a high fixation performance (IQR-span of the horizontal and vertical eye position, M1: 0.23, 0.34 dva; M2: 0.36, 0.49 dva; M3: 0.18, 0.38 dva; M4: 0.37, 0.60 dva), and (2) we averaged across multiple stimulus presentations to obtain robust response profiles. Hence, variations in eye position can have had only minor effects on the present results.”

11) Figure 6B suggests that the neurophysiology data for V4 samples two separate parts of the visual field in the two monkeys.(a) Could the authors state in the paper whether there is in any of their analyses a significant difference in the data from the two monkeys and whether this could have affected the comparison with the MRI?

We ran the underlying analysis separately for M3 and M4 and found the same qualitative pattern of results in both animals. This is now also mentioned in the paper (p11 ln279, p16 ln345). In the comparison with fMRI, V4 differs from V1 since the eccentricity-size relationship of the γ LFP signals are sometimes not significantly different from the MRI results. We also ran the comparison with fMRI separately including either only the electrophysiology pRFs from M3 or M4. The results of these analyses were consistent with the original results based on the data from both animals (p18 ln 367-368).

“Because the visual field coverage in V4 differed between the two animals that were used for electrophysiology, we repeated the analysis in both individuals and observed similar results (not shown).”

(b) Figure 6B suggests that in M4, by far the more eccentric visual field positions were sampled. However, Figure 7B suggests that most of the measured RF sizes are the same or smaller than in M3. Is this correct? And how does this relate to the literature on V4 RF sizes?

We thank the reviewers for catching this inconsistency. The labels for M3 and M4 were flipped in the legend of Figure 7. We corrected this in the revised version of the figure that now makes a more direct comparison between cRF and pRF (see also point 9).